# Effect of metformin on insulin resistance in adults with type 1 diabetes: a 26-week randomized double-blind clinical trial

Jennifer R. Snaith [1,2,3] ✉, Nick Olsen [4], Jennifer Evans[1], Greg M. Kowalski [5], Clinton R. Bruce [6], Dorit Samocha-Bonet [1,2,7], Samuel N. Breit [2,8,9], Deborah J. Holmes-Walker [10,11] & Jerry R. Greenfield[1,2,3]

Insulin resistance is an underrecognized cardiovascular risk factor in type 1 diabetes. The effect of metformin on insulin resistance in adults with type 1 diabetes is unknown. Forty adults with type 1 diabetes, and twenty adults without diabetes were studied in a baseline only cross-sectional study assessing insulin resistance using the two-step hyperinsulinemic-euglycemic clamp. Participants with type 1 diabetes exhibited hepatic (EGP 64% higher), muscle (glucose infusion rate [GIR] 29% lower) and adipose (higher non-esterified fatty acids [NEFA]) insulin resistance. We then conducted a parallel group randomized, placebo-controlled trial to assess the efficacy of metformin 1500 mg ($n = 20$) versus placebo ($n = 20$) in reducing insulin resistance in adults with type 1 diabetes over 26 weeks. The primary outcome was change in endogenous glucose production (EGP) during the low-dose phase of the clamp. Thirty seven of 40 adults with type 1 diabetes completed the study. At 26 weeks, there was no difference in change in EGP between metformin and placebo groups (mean difference 0.2 μmol/kg fat-free mass [FFM]/min [95%CI, −0.4 to 0.8 μmol/kgFFM/min]; $p = 0.53$). There was no increase in hypoglycemia or episodes of ketoacidosis in either group. These results do not support prescribing metformin to reduce hepatic insulin resistance in adults with type 1 diabetes. Australian New Zealand Clinical Trials Registry identifier, ACTRN12619001440112.

Type 1 diabetes is characterized by immune-associated destruction of insulin producing pancreatic beta cells resulting in a life-long reliance on insulin replacement. However, cardiovascular disease remains the main cause of mortality in type 1 diabetes and increased cardiovascular risk is not fully explained by traditional risk factors[1].

Insulin resistance is an important metabolic disturbance in type 1 diabetes. Insulin resistance in type 1 diabetes is considered to be

[1]Clinical Diabetes, Appetite and Metabolism Laboratory, Garvan Institute of Medical Research, Sydney, NSW, Australia. [2]St Vincent's Healthcare Campus, Faculty of Medicine and Health, University of New South Wales Sydney, Sydney, NSW, Australia. [3]Department of Diabetes and Endocrinology, St Vincent's Hospital Sydney, Sydney, NSW, Australia. [4]Stats Central, Mark Wainwright Analytical Centre, University of New South Wales, Sydney, NSW, Australia. [5]Institute for Physical Activity and Nutrition, Metabolic Research Unit, School of Medicine, Deakin University, Geelong, VIC, Australia. [6]Institute for Physical Activity and Nutrition, School of Exercise and Nutrition Sciences, Deakin University, Geelong, VIC, Australia. [7]School of Health Sciences, Faculty of Medicine and Health, University of New South Wales Sydney, Sydney, NSW, Australia. [8]St Vincent's Centre for Applied Medical Research (AMR), Sydney, NSW, Australia. [9]Departments of Clinical Immunology and Immunopathology, St Vincent's Hospital, Sydney, NSW, Australia. [10]Department of Diabetes and Endocrinology, Westmead Hospital, Sydney, NSW, Australia. [11]Westmead Clinical School, Sydney Medical School, The University of Sydney, Sydney, NSW, Australia. ✉e-mail: j.snaith@garvan.org.au

**Table 1 | Baseline demographic and metabolic characteristics of groups with and without type 1 diabetes**

| Characteristic | Type 1 diabetes N = 40 | Control N = 20 |
|---|---|---|
| Age (years) | 37.4 ± 8.8 | 37.0 ± 8.4 |
| Sex (% male) | 60 | 60 |
| BMI (kg/m²) | 26.3 ± 3.8 | 26.2 ± 4.3 |
| WHR | 0.89 ± 0.1 | 0.92 ± 0 1 |
| Men | 0.93 ± 0.1 | 0.96 ± 0.0 |
| Women | 0.82 ± 0.1 | 0.85 ± 0.1 |
| Blood pressure (mmHg) | | |
| Systolic | 119 ± 12 | 117 ± 9 |
| Diastolic | 73 ± 8 | 72 ± 7 |
| Medication use (n, [%]) | | |
| Anti-hypertensives | 4 (10) | 1 (5) |
| Statins | 10 (25) | 0 (0) |
| Fibrates | 0 (0) | 0 (0) |
| Hormonal contraception | 8 (20) | 3 (15) |
| Family history (n, [%]) | | |
| Type 2 diabetes | 8 (20) | 7 (35) |
| Ischaemic heart disease | 5 (12.5) | 5 (25) |
| Ethnicity (% Caucasian) | 40 (100) | 17 (85) |
| Glycemia and diabetes characteristics | | |
| Duration (years) | 22.9 ± 8.9 | NA |
| c-peptide <0.03 nmol/L (n, %) | 38 (95.0) | NA |
| HbA1c (%) | 7.5 ± 0.9 | 5.1 ± 0.3 |
| Insulin units/kg/dayª | 0.6 (0.5, 0.7) | NA |
| Insulin delivery method (n, %) | | |
| MDI | 22 (55) | NA |
| CSII | 18 (45) | NA |
| CGM time in range (%)ª | | |
| <3.0 mmol/L | 0.24 (0, 1.1) | NA |
| <3.9 mmol/L | 2.3 (0.66, 4.4) | NA |
| 3.9–10.0 mmol/L | 58.2 ± 18.8 | NA |
| >10 mmol/L | 38.3 ± 20.1 | NA |
| >13.9 mmol/L | 13.3 (3.2, 23.4) | NA |
| Glucose (mmol/L) | | |
| Mean glucose | 9.5 ± 1.7 | NA |
| Mean overnight glucose (2400–0559) | 9.2 ± 2.2 | NA |
| Mean daytime glucose (0600-2359) | 9.6 ± 1.7 | NA |
| CV (%) | 36.0 ± 6.6 | NA |
| Retinopathy (n, %) | 16 (40.0) | NA |
| Mild non-proliferative | 11 (27.5) | NA |
| Moderate non-proliferative | 5 (12.5) | NA |
| Severe non-proliferative | 0 (0) | NA |
| Proliferative | 0 (0) | NA |
| Maculopathy (n, %) | 7 (17.5) | NA |
| Urine ACR elevated (n, %)ª | 2 (5.4) | NA |

Plus-minus values are mean ± SD. Bracket values are median and IQR, or (n, %) where indicated. Hormonal contraception refers to oral and implanted contraceptives. To convert the values for HbA1c to mmol/mol, multiple the value by 10.93 then subtract 23.5. Elevated ACR defined as ≥2.5 men, ≥3.5 women.

*ACR* albumin-to-creatinine ratio, *BMI* body mass index, *CGM* continuous glucose monitoring, *CSII* continues subcutaneous insulin infusion, *CV* coefficient of variation, *MDI* multiple daily injections, *HbA1c* glycated haemoglobin, *NA* not applicable, *WHR* waist-hip-ratio

ªData available for insulin dose n = 38, CGM n = 37, urine ACR n = 37.

partially driven by the disturbed portal-peripheral insulin concentration gradient created by the subcutaneous route of insulin delivery[2]. Insulin resistance is of interest in type 1 diabetes as studies, including those with surrogate measures of cardiovascular disease, report an association between insulin resistance and cardiovascular risk[3,4].

Whether this association is driven by muscle or liver insulin resistance has not been previously examined[3,5–7]. This differentiation is important, since targeted intervention with adjunctive therapies that ameliorate insulin resistance may provide cardioprotection in type 1 diabetes[8,9].

Metformin, an inexpensive and safe oral medication, is an ideal candidate to address insulin resistance in type 1 diabetes. Although used first-line in the pharmacological management of type 2 diabetes, its mechanisms of action are not entirely understood. In type 2 diabetes, hyperinsulinemic-euglycemic clamp studies demonstrate that metformin alters glucose metabolism at muscle and liver[10]. In addition, metformin has been found to increase circulating growth differentiation factor 15 (GDF15), a stress-regulated hormone with an association with appetite suppression, weight and glucose regulation[11,12]. In type 2 diabetes, large cohort studies, such as the UK Prospective Diabetes Study (UKPDS), indicated an association between metformin and reduced cardiovascular risk and mortality[13]. In type 1 diabetes, systematic reviews of randomized controlled clinical trials demonstrate that metformin reduces total daily insulin dose, often inferring an improvement in insulin resistance[14]. However, studies in type 1 diabetes using the gold-standard clamp technique to directly measure insulin resistance have only been performed in adolescents[15–18]. No study has employed the clamp technique to assess the effect of metformin on insulin resistance in adults with type 1 diabetes.

We designed a cross-sectional study in adults with and without type 1 diabetes to quantify insulin resistance at muscle, liver and adipose tissue; and identify biochemical and clinical features that associated with muscle and liver insulin resistance. We followed this with a 6-month randomized controlled clinical trial, the Insulin Resistance in Type 1 Diabetes Managed with Metformin (INTIMET) study, to test the hypothesis that among adults with type 1 diabetes, relative to placebo, the addition of metformin to insulin would decrease hepatic insulin resistance and improve other cardiometabolic measures, without an increase in adverse effects.

## Results
### Study participants
Men and women with type 1 diabetes for at least 10 years were eligible for the trial if aged between 20 and 55 years, with fasting c-peptide less than 0.3 nmol/L, and an HbA1c less than 9.5% (80 mmol/mol). Twenty adults without diabetes were recruited for baseline studies only. We excluded participants taking medications or with conditions known to impact insulin sensitivity. Complete eligibility criteria are provided in the Methods.

Participants underwent baseline phenotyping assessment between November 2019 and December 2021. In the type 1 diabetes participants, the mean age was 37.4 ± 8.8 years (mean ± SD), with type 1 diabetes duration 22.9 ± 8.9 years, 60% male, BMI 26.3 ± 3.8 kg/m², total daily insulin dose 0.6 (0.5, 0.7) units/kg/day (mean [IQR]) and HbA1c 7.5 ± 0.9% (Table 1). A similar proportion of type 1 diabetes participants used multiple daily insulin injections and insulin pumps. The twenty adults without diabetes were aged 37.0 ± 8.4 years, 60% male, with BMI 26.2 ± 4.3 kg/m² and HbA1c 5.1 ± 0.3%. Therefore, adults with and without type 1 diabetes did not differ in age, sex or anthropometric characteristics (Table 1).

### Baseline analyses in participants with and without type 1 diabetes
**Insulin resistance.** Insulin resistance was assessed at multiple tissue sites, using the three-stage hyperinsulinemic-euglycemic clamp technique with isotope labelled glucose (6,6-²H₂-glucose). The type 1 diabetes group exhibited significantly greater liver, adipose and muscle insulin resistance compared to control participants. Participants with type 1 diabetes had higher endogenous glucose production than controls (EGP; 5.9 ± 2.2 μmol/kg fat free mass [FFM]/min and

3.6 ± 1.7 μmol/kgFFM/min respectively; $p = 0.0002$) and higher non-esterified fatty acids (NEFA) in the low-dose phase (0.08 ± 0.01 mmol/L and 0.02 ± 0.02 mmol/L respectively; $p = 0.001$), indicating impaired insulin-mediated suppression of hepatic glucose release and lipolysis, respectively. Participants with type 1 diabetes had lower glucose infusion rate (GIR) during the high-dose phase (61.9 ± 20.1 μmol/kgFFM/min and 87.7 ± 18.4 μmol/kgFFM/min respectively; $p = 0.00001$), indicating impaired insulin-stimulated glucose uptake by skeletal muscle (Fig. 1). Adjusting for the effect of steady-state insulin and glucose concentrations on clamp variables did not change this result (Supplementary Table 1).

**Metabolic characteristics.** Comprehensive cross-sectional metabolic assessments were performed at baseline for all participants with and without diabetes, and at 26 weeks for randomized participants with type 1 diabetes.

At baseline, augmentation index (AIx; a measure of arterial stiffness and validated surrogate of cardiovascular disease) was higher in participants with type 1 diabetes than participants without diabetes (Table 2)[19]. Serum levels of soluble intercellular adhesion molecule 1 (sICAM-1) and GDF15 were also higher in participants with type 1 diabetes (Table 2). Despite greater insulin resistance in type 1 diabetes, serum adiponectin and body composition measurements, including visceral adipose tissue (VAT) mass, were similar (Table 2). Uric acid and insulin-like growth factor-1 (IGF-1) were lower, and sex hormone binding globulin (SHBG) was higher, in participants with type 1 diabetes (Table 2). Triglycerides were lower in type 1 diabetes individuals but not after adjusting for statin use ($p = 0.061$). The type 1 diabetes group had higher high-density lipoprotein (HDL) cholesterol concentrations after adjusting for statin use (1.4 ± 0.4 vs 1.2 ± 0.2 mmol/L, respectively, $p = 0.044$).

**Metabolic characteristics and relationships with insulin resistance.** To examine the phenotype of insulin resistance, relationships between cardiometabolic measures and sites of insulin resistance were assessed using baseline measures. In the total cohort, liver insulin resistance did not relate to any metabolic variable except sICAM-1, soluble endothelial selectin (sE-selectin) and alkaline phosphatase (ALP) (Supplementary Table S2). Muscle insulin resistance (i.e. 1/GIR) positively associated with Log-AIx ($r = 0.258$; $p = 0.048$), BMI, VAT mass, serum inflammatory markers (sICAM-1, sE-selectin), liver enzymes (ALT, GGT, ALP) and LDL cholesterol (Supplementary Table S2). In participants with type 1 diabetes, muscle insulin resistance additionally correlated directly with HbA1c, insulin dose, mean overnight glucose, and inversely with glycemic variability (Supplementary Table S2). Adjustment for HbA1c, but not fasting glucose, attenuated the difference in GIR between participants with and without type 1 diabetes (Supplementary Table S3).

**Efficacy analyses – metformin in type 1 diabetes**
The baseline characteristics of participants with type 1 diabetes randomized to metformin or placebo were similar, except for a lower total daily insulin dose in the metformin group (Table 3). After completing baseline studies, participants with type 1 diabetes were randomized to metformin extended release, or identical appearing placebo, as 500 mg tablets, titrated from 500 mg for 7 days, then 1000 mg for 7 days then 1500 mg for the remaining 26-week treatment period. Dosage could be adjusted to accommodate adverse effects, and a submaximal dose was permitted. Thirty-seven of 40 participants completed the 26-week intervention period, with no pattern to the participant dropout by treatment group (Fig. 2). Medication adherence assessed by tablet count was similar between groups (89 vs 88% metformin vs placebo; $p = 0.88$).

**Insulin resistance.** The primary endpoint was change in liver insulin sensitivity at 26 weeks. Treatment with metformin did not significantly alter EGP (treatment by time interaction: $\beta = 0.2$ [95% CI −0.4 to 0.8],

$p = 0.53$) compared to placebo after 26 weeks (Fig. 3, Supplementary Table S4).

Pre-specified secondary endpoints included the change in muscle and adipose insulin sensitivity at 26 weeks. The change in GIR in the metformin group was 8.3 μmol/kgFFM/min (95% CI 0.8–15.9; $p = 0.03$), and 6.6 μmol/kgFFM/min (−1.2 to 14.6; $p = 0.10$) in the placebo group, with no overall difference in GIR at 26 weeks between groups (GIR treatment by time interaction: $\beta = 1.6$ [95% CI −4.8 to 7.9], $p = 0.62$). Additionally, there was no significant difference in NEFA from baseline to 26 weeks between groups (treatment by time interaction: $\beta_{logscale} = -0.12$ [95% CI −0.5 to 0.3], $p = 0.54$). Adjustment for age, sex, baseline BMI, baseline HbA1c or insulin delivery method did not change the result for insulin resistance measures (Supplementary Table S5).

**Total daily insulin, glycemia and weight.** The results for the pre-specified secondary endpoint of change in insulin dose at 26 weeks are shown in Fig. 4. Metformin significantly reduced total daily insulin dose relative to placebo (estimated treatment difference [ETD] −0.10 units/kg/day [95% CI −0.15 to −0.04], $p = 0.0008$; metformin −0.03 units/kg/day [95% CI −0.08 to 0.030], placebo +0.04 units/kg/day [95% CI −0.01 to 0.10]), and the result remained significant after adjustment for multiple comparisons ($p = 0.02$). Adjustment for age, sex, baseline BMI, baseline HbA1c or insulin delivery method did not change the result for total daily insulin dose (Supplementary Table S5).

HbA1c and continuous glucose monitoring parameters were used to assess the effect of metformin on glycemia. Participants with type 1 diabetes wore blinded continuous glucose monitoring (CGM) for up to 2 weeks prior to the baseline and 26-week study visits. The first 20 enrolled participants were provided the Medtronic iPro ® Enlite system, (Medtronic Northridge, CA, changed after 7 days, in total 2 devices worn per participant) and the final twenty participants were provided the Dexcom G6 monitor (Dexcom, San Diego, CA) due to discontinued supply of the Medtronic system mid-trial. There were no differences between the trial groups for HbA1c, % time within target range (3.9–10 mmol/L [70–180 mg/dL]), % time below range, <3.9 mmol/L [<70 mg/dL], time below range, <3.0 mmol/L [<54 mg/dL]), % time above range >10.0 mmol/L [>180 mg/dl]), % time above range >13.9 mmol/L (>250 mg/dl), or glycemic variability (coefficient of variation) over 26 weeks (Supplementary Tables S4, S6).

**Other secondary endpoints and exploratory analyses.** After 26 weeks, participants randomized to metformin had lower VAT (ETD $\beta_{logscale}$ −0.3 [95% CI −0.7 to −0.02], $p = 0.04$; metformin −15.2 g [95% CI −88.6 to 58.2], placebo +94.2 g [95% CI −208 to 397]) and higher serum GDF15 levels (ETD $\beta_{logscale}$ 0.3 [95% CI 0.1–0.4], $p = 0.001$; metformin +399 pg/mL [95% CI [93–704], placebo +17 pg/mL [95% CI −102 to 136] compared to placebo (Supplementary Table S4 and Fig. 5). The $p$-value for GDF15 remained significant after accounting for multiple comparisons ($p = 0.03$), although the reduction in VAT was no longer significant.

There were no significant differences between the trial groups in AIx, weight or other body composition, biochemical or anthropometric secondary endpoints at 26 weeks (full detail in Supplementary Table S4). Microbiome analyses, a pre-specified secondary endpoint, is not reported in this manuscript.

To further examine contributors to the effect of metformin on insulin dose, exploratory analysis of the impact of metformin on glucagon and caloric intake was performed. Metformin did not alter fasting glucagon levels or glucagon response to insulin during the clamp (Supplementary Table S7). Participants kept detailed diet records for up to 14 days prior to their baseline and 26-week study visits. There was no difference in total caloric, carbohydrate, protein, fat or sugar intake between groups at 26 weeks (Supplementary Table S7).

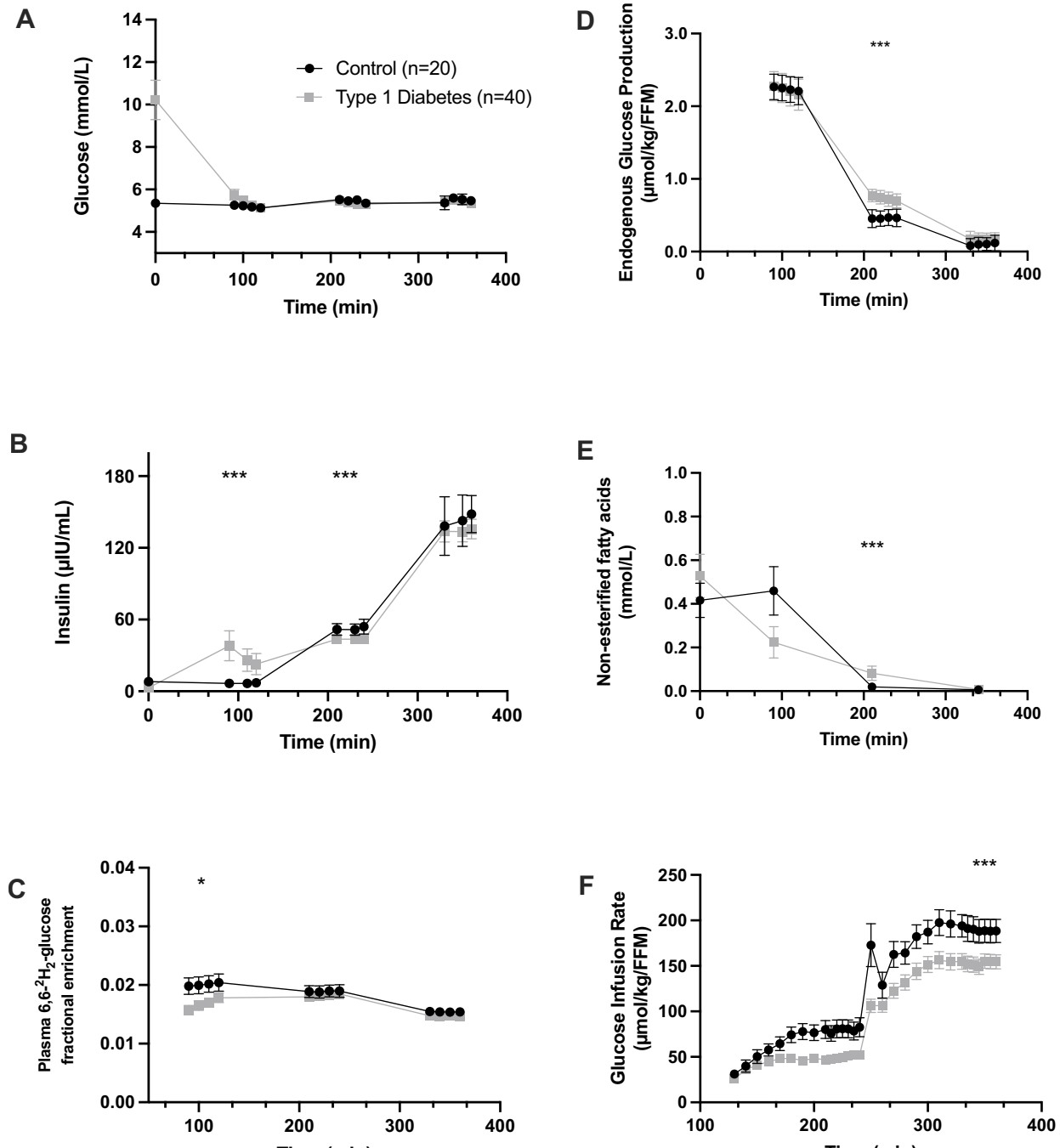

**Fig. 1 | Clamp measurements from the hyperinsulinemic-euglycemic clamp in type 1 diabetes and control groups, displayed across the time course of the clamp. A** Though participants with type 1 diabetes arrived with hyperglycaemia, steady state glucose levels were equivalent between type 1 diabetes and control participants across all clamp phases. **B** Steady state plasma insulin concentration was higher in type 1 diabetes participants in the basal phase ($p = 0.00001$), lower in the low-dose phase ($p = 0.001$) and similar in the high-dose phase ($p = 0.55$). The higher basal-phase insulin concentrations in the type 1 diabetes group reflect the infused insulin necessary to achieve euglycemia. **C** Plasma 6,6-$^2$H$_2$-glucose isotope enrichment was lower in the type 1 diabetes group in the basal phase ($p = 0.00002$), and equivalent between groups in the low and high-dose phases. **D** Endogenous glucose production (EGP) decreased across the stages of the clamp indicating dose-dependent suppression in

response to insulin. EGP during the low-dose step of the clamp was 64% higher in type 1 diabetes indicating impaired suppression of hepatic endogenous glucose production ($p = 0.0002$). **E** Non-esterified fatty acid (NEFA) levels during the low-dose step of the clamp were higher in type 1 diabetes indicating impaired suppression of lipolysis by insulin ($p = 0.001$). **F** Glucose infusion rate (GIR) requirements increased across the course of the clamp and was steady by the sampling period. The GIR during the high-dose step of the clamp was 29% lower in type 1 diabetes, indicating muscle insulin resistance ($p = 0.00001$). Together these graphs show the presence of multi-tissue insulin resistance in type 1 diabetes relative to control participants. Analyses are two-sided, using t-tests or Mann–Whitney U according to data distribution, without adjustment for multiple testing. Data are presented as means and 95% confidence intervals. *$p < 0.05$, **$p < 0.01$, ***$p < 0.001$.

**Table 2 | Biochemical, body composition and vascular characteristics of the groups with and without type 1 diabetes**

| Characteristic | Type 1 diabetes N = 40 | Control N = 20 | p-value |
|---|---|---|---|
| **Inflammatory and vascular** | | | |
| AIx (%)[a] | 13.0 (7.6, 18.8) | 4.3 (−4.0, 15.4) | 0.046 |
| sICAM-1 (ng/mL) | 213.9 ± 47.7 | 180.9 ± 39.8 | 0.010 |
| sE-selectin (ng/mL) | 35.0 ± 14.2 | 28.0 ± 10.1 | 0.05 |
| IL-6 (pg/mL)[a] | 1.0 (0.7, 1.5) | 0.9 (0.5, 1.1) | 0.05 |
| **Body composition** | | | |
| **DXA** | | | |
| total fat mass (kg) | 24.1 ± 9.2 | 24.2 ± 8.8 | 0.98 |
| total fat free mass (kg) | 55.4 ± 9.6 | 54.0 ± 10.4 | 0.59 |
| VAT (kg) | 0.48 (0.2, 0.9) | 0.58 (0.1, 1.0) | 0.72 |
| **MRI[b]** | | | |
| Mean anterior thigh MFI (%) | 4.0 ± 1.0 | 4.5 ± 1.5 | 0.28 |
| ASAT volume (L) | 5.1 (3.2, 7.8) | 8.1 (3.8, 10.5) | 0.19 |
| Liver fat (%) | 1.3 (0.9, 2.0) | 1.5 (0.8, 4.3) | 0.86 |
| **Transient elastography** | | | |
| CAP score (dB/m) | 230 ± 37 | 232 ± 36 | 0.88 |
| **Liver function and biochemistry** | | | |
| Bilirubin (µmol/L) | 9 (8, 14) | 9 (6, 13) | 0.29 |
| Albumin (g/L) | 37 ± 2 | 39 ± 3 | 0.03 |
| ALT (U/L) | 26 ± 13 | 25 ± 12 | 0.79 |
| AST (U/L) | 20 ± 10 | 21 ± 6 | 0.61 |
| GGT (U/L) | 15 (12, 29) | 15 (11, 19) | 0.37 |
| ALP (U/L) | 70 ± 26 | 58 ± 19 | 0.03 |
| Uric acid (mmol/L) | 0.25 ± 0.07 | 0.31 ± 0.07 | 0.002 |
| **Lipids** | | | |
| Total cholesterol (mmol/L) | 4.3 ± 0.8 | 4.4 ± 0.7 | 0.47 |
| LDL (mmol/L) | 2.6 ± 0.6 | 2.8 ± 0.6 | 0.27 |
| HDL (mmol/L) | 1.3 ± 0.4 | 1.2 ± 0.2 | 0.09 |
| Cholesterol/HDL ratio | 3.3 ± 0.7 | 3.7 ± 0.7 | 0.05 |
| Non-HDL (mmol/L) | 2.9 ± 0.7 | 3.2 ± 0.7 | 0.12 |
| Triglycerides (mmol/L) | 0.8 ± 0.4 | 1.1 ± 0.3 | 0.02 |
| **Liver metabolites** | | | |
| IGF-1 (nmol/L) | 18.9 ± 4.7 | 24.7 ± 5.5 | 0.0001 |
| SHBG (nmol/L) | 53 (36, 70) | 36 (23, 47) | 0.003 |
| Men | 49 (35, 58) | 27 (23, 40) | 0.003 |
| Women[c] | 74 (62, 113) | 45 (32, 77) | 0.09 |
| Adiponectin (µg/mL)[a] | 4.9 (3.1, 10.5) | 4.9 (3.2, 7.7) | 0.53 |
| GDF15[a] (pg/mL) | 593 (472, 676) | 493 (432, 618) | 0.04 |

Plus-minus values are mean ± SD. Bracket values are median and interquartile range. *P*-values were obtained using two-sampled *T*-tests or Mann–Whitney U-tests and without adjustment for multiple testing.

*AIx* augmentation index, *ALP* alkaline phosphatase, *ALT* alanine transaminase, *ASAT* abdominal subcutaneous adipose tissue, *AST* aspartate transaminase, *CAP* controlled attenuation parameter, *DXA* dual-energy x-ray absorptiometry, *GDF15* growth differentiation factor 15, *GGT* gamma-glutamyl transferase, *HDL* high density lipoprotein, *IGF-1* insulin like growth factor-1, *IL-6* interleukin 6, *LDL* low density lipoprotein, *MFI* muscle fat infiltration, *MRI* magnetic resonance imaging, *NA* not applicable, *sE-selectin* soluble E-selectin, *sICAM1* soluble intercellular adhesion molecule 1, *SHBG* sex hormone binding globulin, *VAT* visceral adipose tissue

[a]Log transformed for analysis.

[b]MRI controls n = 11, type 1 diabetes n = 24. IGF-1 controls n = 19, type 1 diabetes n = 40. GDF15 controls n = 20, type 1 diabetes n = 39 (single case excluded).

[c]Excluded participants taken hormone-based contraceptives (after excluding these participants, data available SHBG controls n = 17, type 1 diabetes n = 32).

## Safety

Overall, metformin was well tolerated, with an acceptable side effect profile (Supplementary Tables S6, S8). One participant in the metformin group (5%) and two in the placebo group (10%) discontinued treatment prior to the 26 week follow up visit (Fig. 2). There was no pattern to the participant dropout by treatment group. The participant in the metformin group discontinued 22 weeks after randomization due to pregnancy. In the placebo group, one participant developed intolerable gastrointestinal side effects (bloating), and the other elected to discontinue due to COVID-19 related lockdowns impacting personal capacity to complete the study (Fig. 2). The placebo group had more frequent gastrointestinal adverse effects (nausea, bloating) and required dose de-escalation in one individual and led to study drop out in another (Supplementary Table S8). Gastrointestinal side effects were absent in the metformin group. In the metformin group, the two hospitalization events occurred in the same individual and was considered unrelated to the study drug (appendicitis, and pre-syncope leading to emergency department presentation without admission). Iron deficiency was considered likely secondary to heavy menstrual periods. There were no episodes of diabetic ketoacidosis or severe hypoglycemia in either group. The frequency of hypoglycemia (% time <3.9 mmol/L or <3.0 mmol/L) occurred in similar frequency in metformin and placebo groups (Supplementary Table S6). Hypoglycemia was not defined by the American Diabetes Association levels of hypoglycemia due to the trial design and conduct preceding the introduction of these definitions.

## Discussion

This study, the first randomized controlled clinical trial assessing the effect of adjunctive metformin on insulin resistance in adults with type 1 diabetes, did not demonstrate improvement in liver, muscle or adipose tissue insulin resistance after treatment for 26 weeks. These results do not support the use of metformin to reduce insulin resistance in adults with type 1 diabetes. This finding contrasts with studies of metformin in adolescents with type 1 diabetes, that reported improved peripheral insulin sensitivity and no change in hepatic glucose production (a measure of liver insulin resistance)[15,16]. The influence of pubertal factors on insulin resistance in adolescents may explain this difference in results[20].

Although there was no evidence of an effect on measures of insulin resistance, metformin was associated with a reduction in total daily insulin dose by 0.1 units/kg/day relative to the placebo group. The finding of an insulin dose reduction mirrors systematic review findings pooling studies with insulin dose as a trial endpoint performed in both adults and youth with type 1 diabetes[14]. Insulin dose reduction is often assumed to imply a reduction in insulin resistance. However, no prior studies have employed the clamp technique to directly measure insulin resistance in adults. Our findings suggest that in adults with type 1 diabetes, metformin may have metabolic impacts that translate to sparing of insulin requirements via effects independent of insulin resistance.

The mechanisms for the reduction of insulin dose in the absence of change in insulin resistance is of interest. Since carbohydrate intake and glucagon levels did not change with metformin treatment, we could not attribute the effect to appetite suppression, or to glucagon impacting the net effect of metformin on glucose metabolism. There is mounting evidence for an interaction between metformin and the gut, a phenomena that is clearly apparent in metformin exposed individuals that exhibit increased colonic and small intestinal glucose tracer uptake when undergoing fluoro-deoxyglucose (FDG)-positron emission topography (PET) imaging[21–23]. The mechanism by which metformin increases intestinal glucose uptake is thought to be due to increased glucose transporter (GLUT) and reduced SGLT1 apical expression in enterocytes[24]. Other postulated theories for metformin's action on the gut include effects on lactate-glucose cycling, bile acid absorption, incretin secretion, GDF15 and changes to the gut microbiome[21]. Our group will report the effect of metformin on the microbiome in type 1 diabetes in a separate manuscript.

GDF15 is a novel stress-responsive cytokine that is gaining interest as a metabolism biomarker and a potential therapeutic target. GDF15

**Table 3 | Baseline characteristics of participants with type 1 diabetes by treatment group**

| Characteristic | Metformin (N = 20) | Placebo (N = 20) |
|---|---|---|
| Age (year) | 38.1 ± 7.1 | 36.7 ± 10.4 |
| Sex (% male) | 60 | 60 |
| BMI (kg/m²) | 27.0 ± 3.7 | 25.8 ± 3.8 |
| WHR | 0.88 ± 0.1 | 0.90 ± 0.1 |
| Men | 0.93 ± 0.1 | 0.93 ± 0.1 |
| Women | 0.81 ± 0.1 | 0.83 ± 0.0 |
| Blood pressure (mmHg) | | |
| Systolic | 117 ± 13 | 121 ± 12 |
| Diastolic | 73 ± 7 | 73 ± 8 |
| Medication use (n, [%]) | | |
| Anti-hypertensive | 1 (5) | 3 (15) |
| Statin | 5 (25) | 5 (25) |
| Hormonal contraception | 3 (15) | 5 (25) |
| Family history (n, [%]) | | |
| Type 2 diabetes | 3 (15) | 5 (25) |
| Ischaemic heart disease | 3 (15) | 2 (10) |
| Ethnicity (% Caucasian) | 20 (100) | 20 (100) |
| Glycemia and insulin | | |
| HbA1c (%) | 7.4 ± 0.8 | 7.6 ± 1.1 |
| Insulin units/kg/day[a,b] | 0.54 (0.47, 0.64) | 0.66 (0.52, 0.93) |
| Insulin delivery method (n, %) | | |
| MDI | 11 (55) | 11 (55) |
| CSII | 9 (45) | 9 (45) |
| CGM time in range (%)[b] | | |
| <3.0 mmol/L (<55 mg/dL) | 0.3 (0.0, 1.7) | 0.2 (0.0, 1.1) |
| <3.9 mmol/L (<70 mg/dL) | 2.5 (0.8, 6.3) | 1.5 (0.6, 3.9) |
| 3.9–10.0 mmol/L (70–180 mg/dL) | 58.5 ± 17.9 | 58.0 ± 20.5 |
| >10 mmol/L (>180 mg/dL) | 37.4 ± 19.1 | 39.4 ± 21.8 |
| >13.9 mmol/L (>250 mg/dL) | 15.0 (3.0, 20.2) | 12.5 (4.0, 26.0) |
| Mean glucose (mmol/L) | 9.3 ± 1.6 | 9.6 ± 2.0 |
| Mean overnight glucose (2400–0600, mmol/L) | 8.9 ± 2.1 | 9.4 ± 2.3 |
| Mean daytime glucose (0600-2400, mmol/L) | 9.5 ± 1.6 | 9.7 ± 1.9 |
| CV (%) | 36.6 ± 7.4 | 35.4 ± 5.6 |
| Diabetes | | |
| Duration (years) | 24.2 ± 8.5 | 21.7 ± 9.3 |
| Retinopathy (n, %)[b] | 8 (40.0) | 8 (40.0) |
| Mild non-proliferative | 7 (35.0) | 4 (20.0) |
| Moderate non-proliferative | 1 (5.0) | 4 (20.0) |
| Severe non-proliferative | 0 (0.0) | 0 (0.0) |
| Proliferative | 0 (0.0) | 0 (0.0) |
| Maculopathy (n, %) | 2 (20) | 3 (15) |
| Urine ACR elevated[b] (n,%) | 1 (5.6) | 1 (5.5) |
| c-peptide <0.03 nmol/L (n, %) | 19 (95) | 19 (95) |
| Body composition | | |
| DEXA | | |
| Total fat mass (kg) | 25.5 ± 10.7 | 22.6 ± 7.3 |
| Total fat free mass (kg) | 55.8 ± 9.4 | 55.1 ± 10.0 |
| Visceral adipose tissue (kg) | 0.4 (0.2, 0.7) | 0.5 (0.1, 1.1) |
| MRI[b] | | |
| Mean anterior thigh muscle fat infiltration (%) | 4.0 ± 1.3 | 4.0 ± 0.8 |

**Table 3 (continued) | Baseline characteristics of participants with type 1 diabetes by treatment group**

| Characteristic | Metformin (N = 20) | Placebo (N = 20) |
|---|---|---|
| Abdominal subcutaneous adipose tissue volume (L) | 7.3 (3.9, 10.4) | 4.4 (3.1, 6.4) |
| Liver fat (%) | 1.3 (0.9, 1.5) | 1.9 (0.9, 2.8) |
| Transient elastography | | |
| CAP score (dB/m) | 222 ± 31 | 238 ± 41 |
| Blood metabolites | | |
| Total cholesterol (mmol/L) | 4.4 ± 0.8 | 4.2 ± 0.8 |
| LDL (mmol/L) | 2.7 ± 0.7 | 2.5 ± 0.6 |
| HDL (mmol/L) | 1.3 ± 0.3 | 1.4 ± 0.5 |
| Cholesterol/HDL ratio | 3.4 ± 0.6 | 3.3 ± 0.8 |
| Non-HDL (mmol/L) | 3.0 ± 0.7 | 2.9 ± 0.6 |
| Triglycerides (mmol/L) | 0.8 ± 0.4 | 0.8 ± 0.4 |
| Adiponectin (µg/mL) | 4.2 (3.4, 13.0) | 5.6 (2.5, 9.3) |
| GDF15* (pg/mL) | 553 (471, 675) | 642 (474, 688) |
| Inflammatory and vascular | | |
| Augmentation index (AIx, %) | 10.8 (7.1, 16.5) | 13.0 (8.1, 21.6) |
| sICAM-1 (ng/mL) | 214.4 ± 47.5 | 213.4 ± 49.1 |
| sE-selectin (ng/mL) | 34.6 ± 14.4 | 35.4 ± 14.5 |
| IL-6 (pg/mL) | 1.1 (0.7, 1.7) | 1.0 (0.7, 1.5) |

Values are mean ± SD. Bracket values are median and interquartile range. Hormonal contraception refers to oral and implanted contraceptives. Elevated albumin-to-creatinine ratio defined as ≥2.5 men, ≥3.5 women. To convert the values for HbA1c to mmol/mol, multiple the value by 10.93 then subtract 23.5.
*ACR* Albumin-to-creatinine ratio, *CGM* continuous glucose monitoring, *CSII* continues sub-cutaneous insulin infusion, *CV* coefficient of variation, *GDF15* growth/differentiation factor-15, *HDL* high density lipoprotein, *IL-6* interleukin 6, *LDL* low density lipoprotein, *MDI* multiple daily injection, *sE-selectin* soluble endothelial selectin, *sICAM1* soluble intercellular adhesion molecule 1.
[a]There was an imbalance in total daily insulin dose between groups (t-test p = 0.02).
[b]Data available for insulin dose metformin n = 19, placebo n = 19, CGM metformin n = 20, placebo n = 17, urine ACR metformin n = 18, placebo n = 19, high-dose insulin phase metformin n = 20, placebo n = 19, MRI metformin n = 11, placebo n = 12, GDF15 metformin n = 19, placebo n = 20.

was elevated at baseline in the type 1 diabetes group. This finding is consistent with the expectation that GDF15 is elevated in diabetes but likely offers a protective function. Therefore, paradoxically, whilst an elevated basal GDF15 level may represent poorer health, increased levels are thought to be protective, and pharmacologically inducing GDF15 has been shown to be metabolically favourable.

Metformin led to a significant increase in serum GDF15 levels compared to placebo. This is consistent with prior human clinical studies in prediabetes, type 2 diabetes and obesity[11,25–29]. Data from rodents and primates demonstrated that GDF15 treatment induces reductions in food intake, body weight, triglyceride concentrations and implicates GDF15 in metformin's glucoregulatory actions[12,30,31]. To our knowledge, no prior study has examined the effect of metformin on GDF15 in type 1 diabetes, nor in humans with diabetes in a placebo-controlled setting. In this study, it is unclear whether GDF15 is mechanistically related to insulin dosage, and/or glucose metabolism. We hypothesize that GDF15 may promote metformin-induced glucose transport into enterocytes, thereby creating an intestinal glucose sink and reducing insulin dose requirements. Animal studies would assist in testing this hypothesis.

Insulin monotherapy remains the mainstay of treatment of type 1 diabetes, yet is associated with iatrogenic insulin resistance and insulin resistance is indirectly associated with increased CV risk[2,4]. Using the hyperinsulinemic-euglycemic clamp technique with glucose isotope, we demonstrated that participants with type 1 diabetes had considerable muscle (29% lower GIR), liver (64% higher EGP) and adipose tissue insulin resistance, and greater arterial stiffness, than control

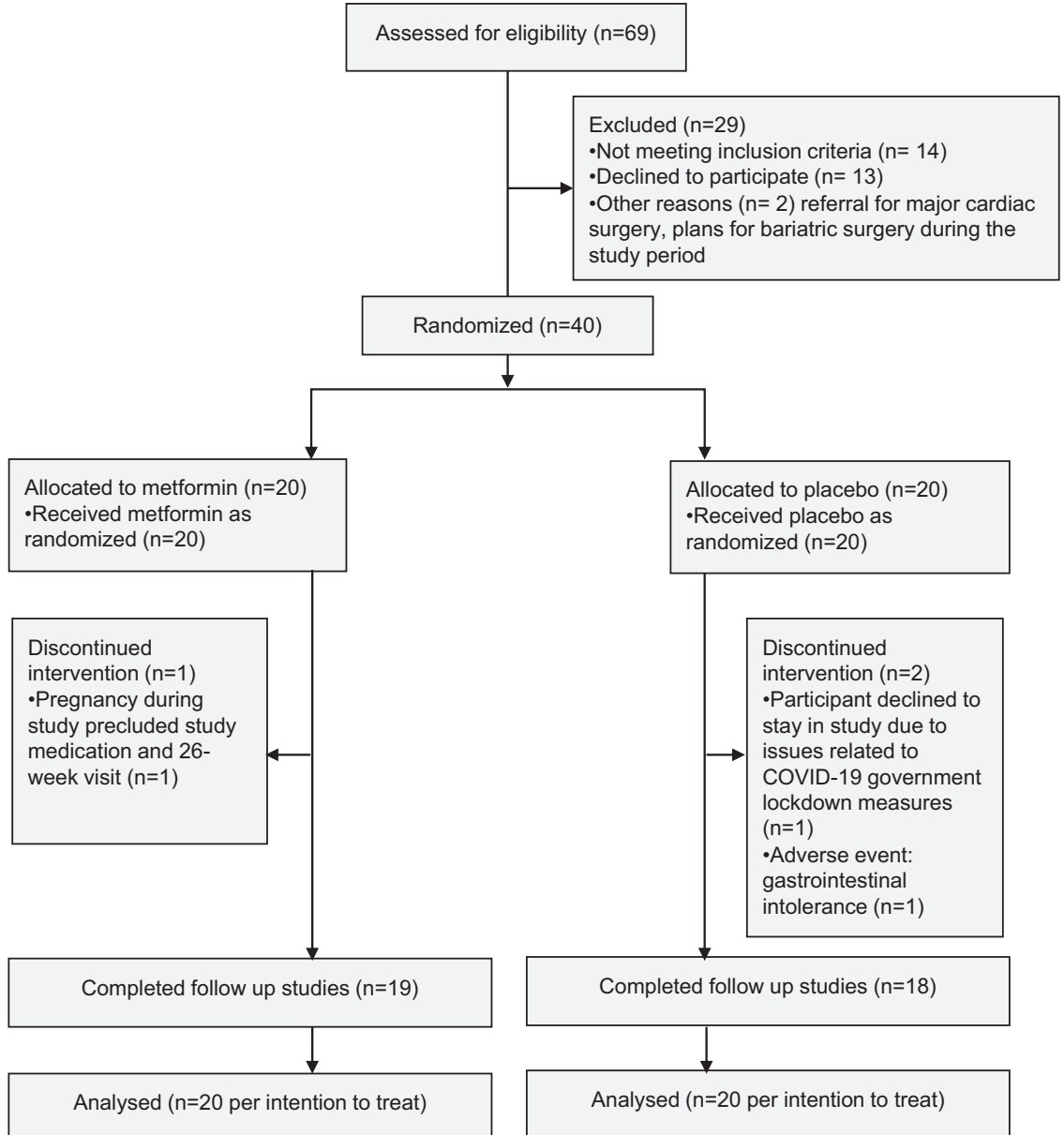

**Fig. 2 | Screening, randomization and follow up sample sizes.** Between November 2019 and December 2021, 40 adults with type 1 diabetes were enroled in the study, then randomized 1:1 to metformin or placebo. Nineteen participants in the metformin group, and 18 in the placebo group completed the study and analyses were conducted by intention to treat.

participants. Muscle insulin resistance was associated with arterial stiffness, suggesting that muscle insulin resistance may be a key therapeutic target to reduce vascular risk in type 1 diabetes. From studies in type 2 diabetes, it has been proposed that metformin primarily inhibits hepatic glucose output (possibly via AMP kinase activation), secondarily reducing fatty acid oxidation, with indirect effects on skeletal muscle glucose disposal[10,32]. This provided rationale for the investigation of metformin in type 1 diabetes.

Our study suggests that in type 1 diabetes, muscle insulin resistance is influenced by hyperglycaemia. It is also likely induced by the direct absorption of subcutaneously administered insulin into the peripheral circulation, bypassing hepatic insulin clearance and distorting the usual balance between portal and peripheral insulin concentrations[2,33]. Exposure to chronic hyperinsulinemia may disrupt insulin transduction and is associated with defective oxidative glucose disposal, endothelial dysfunction and obesity in type 1 diabetes[2,34,35]. Thus, reducing insulin resistance, hyperinsulinemia, and insulin dose

may all be important therapeutic targets in the treatment of type 1 diabetes.

However, it has not yet been established whether reducing total daily insulin dose may lead to a reduction in insulin resistance in type 1 diabetes. Thus, the clinical relevance of our finding of insulin dose reduction is uncertain. Notably, in a Diabetes Control and Complications Trial/Epidemiology of Diabetes Interventions and Complications (DCCT/EDIC) follow up study, higher insulin doses were adversely associated with cardiovascular risk factors, and each 0.1 units/kg/day insulin dose increase predicted a 6% increase in cardiovascular disease risk over 30 years[36]. Although the association with cardiovascular outcomes was attenuated after adjustment for traditional risk factors, the findings raise the hypothesis that minimizing insulin dose may have beneficial consequences for cardiovascular risk over time[36]. Indeed lower insulin dose requirements is one of the proposed mechanisms by which insulin pump therapy has been linked with lower mortality compared to multiple daily injections[37].

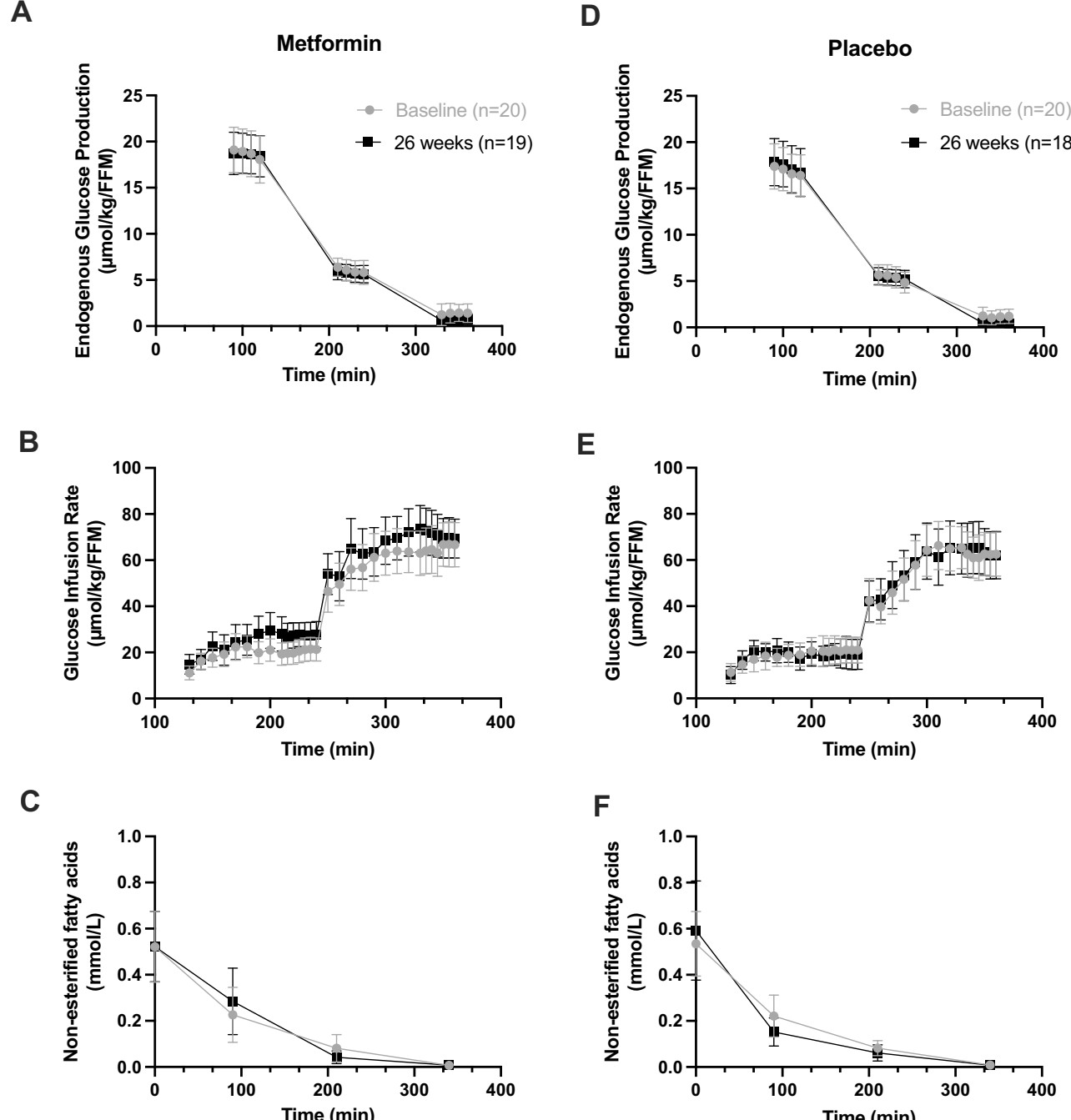

**Fig. 3 | Insulin resistance measures at baseline and after 26 weeks by treatment group.** The graphs show the pre- and post-treatment endogenous glucose production (Panel **A**), glucose infusion rate (Panel **B**) and non-esterified fatty acids (Panel **C**) in the participants with type 1 diabetes treated with metformin or placebo (Panel **D**, **E** and **F**). Metformin had no significant effect on measures of liver, muscle or adipose insulin resistance. Analyses are two-sided using generalized linear mixed modelling without adjustment for multiple testing. Data are presented as means and 95% confidence intervals.

Furthermore, though metformin failed to reduce insulin resistance in type 1 diabetes, this finding does not discount possible pleotropic effects to reduce cardiovascular risk in this population. The REMOVAL study, an international multicentre trial in adults age 40 years or older with multiple cardiovascular risk factors randomized to metformin 1000 mg twice daily or placebo, sought to provide cardiovascular data for metformin in type 1 diabetes, using carotid artery intima-media thickness (cIMT), a validated surrogate measure of atherosclerotic cardiovascular disease[38,39]. Metformin did not significantly reduce the primary endpoint, mean far wall cIMT, although a significant reduction was seen a pre-specified subgroup analysis in

participants that had never-smoked[40]. In REMOVAL, there were mixed results regarding insulin dose reduction with metformin, with a significant reduction seen from 6 months after randomization. The mechanism for metformin offering possible CV protection in type 1 diabetes remains uncertain, and whether the mechanism involves insulin dose conservation requires further study[38].

The clinical significance of, and the mechanisms driving, hepatic insulin resistance in type 1 diabetes are uncertain. In our study, as liver fat was not higher, and did not correlate with liver insulin resistance, liver fat was considered unlikely to contribute to the development of liver insulin resistance in type 1 diabetes. This is unlike the

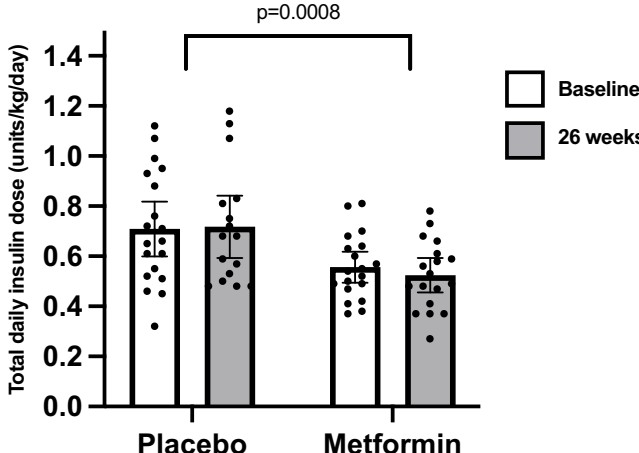

**Fig. 4 | Insulin dose at baseline and after 26 weeks by treatment group.** Metformin significantly decreased total daily insulin dose (estimated treatment difference −0.1 units/kg/day [95% CI −0.15 to −0.04]; *p* = 0.0008). Analyses are two-sided using generalized linear mixed modelling without adjustment for multiple testing. Data are presented raw as means and 95% confidence intervals and were available for *n* = 19 placebo and *n* = 19 metformin at baseline, *n* = 16 placebo and *n* = 18 metformin at 26 weeks.

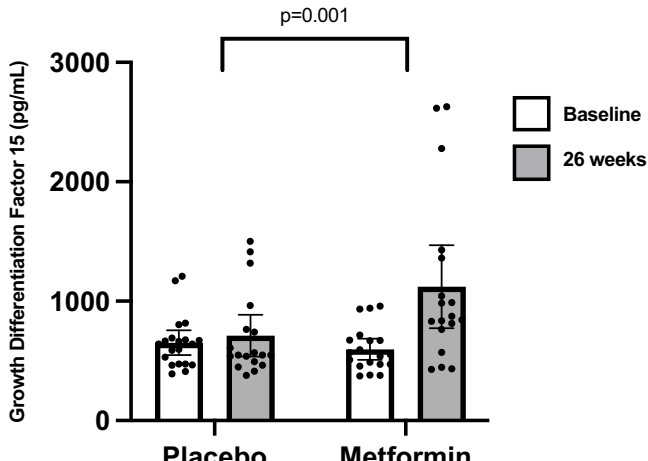

**Fig. 5 | GDF15 at baseline and after 26 weeks by treatment group.** Metformin significantly increased growth differentiation factor 15 (GDF15) compared to placebo (estimated treatment difference 382 pg/mL [95% CI 59–704]; *p* = 0.001). Analyses are two-sided using generalized linear mixed modelling without adjustment for multiple testing. Data are presented raw as means and 95% confidence intervals and were available for *n* = 20 placebo and *n* = 19 metformin at baseline, *n* = 18 placebo and *n* = 18 metformin at 26 weeks.

well-established relationship between hepatic fat and insulin resistance reported in some studies of type 2 diabetes[41]. Impaired hepatic insulin sensitivity may also arise due to hyperglucagonemia causing increased EGP, an underappreciated defect in type 1 diabetes[42]. Glucagon itself can exacerbate post-prandial hyperglycaemia, insulin resistance and cardiovascular risk[42,43]. However, we did not demonstrate elevated glucagon levels relative to participants without diabetes. Further, impaired suppression of lipolysis may increase gluconeogenic substrate flux (such as fatty acids or glycerol) from peripheral adipose tissue to the liver which then interfere with insulin sensitivity at the hepatic site[44]. Thus, in type 1 diabetes, apparent hepatic insulin resistance may in fact be a downstream consequence of peripheral insulin resistance.

A major strength of our study was that we included individuals across a wide range of insulin sensitivities. This distinguishes our study from other type 1 diabetes studies, which included individuals with overweight or obesity, high insulin dose requirements or metabolic syndrome features[15,16,45,46]. This is important given our demonstration of prominent insulin resistance in type 1 diabetes without an elevated BMI or visceral adiposity and offers the benefit of generalizability across the BMI spectrum. This benefit is also a limitation. Our trial could have provided different results if targeted to a cohort with a higher BMI, or higher HbA1c. Our enrolment criteria excluded those with HbA1c greater than 9.5%, a population in whom glucotoxicity may significantly contribute to insulin resistance. However, even with this restriction, HbA1c appeared to predict insulin resistance in our cohort. Other limitations include our predominantly Caucasian cohort which limits generalizability. Our study may have been underpowered to detect the effect of metformin on GIR, or other secondary outcomes, and differences in clamp methodology may account for differences compared to the findings reported in adolescents[15,16]. Although EGP was almost completely suppressed in most participants, we did not correct for tracer estimated EGP in the high-dose phase. This may have led to an underestimation of GIR in some participants, however there were no observable differences in between group EGP between participants with and without diabetes, or between placebo and metformin groups. Notably insulin dose adjustment was left to the discretion of the participant with diabetes and their treating clinician. Therefore, we cannot determine whether the reduction in insulin dose impacted or masked glycemic efficacy measures such as HbA1c. We only assessed EGP in the fasting state. The impact of metformin on prandial glucose and glucagon kinetics were not assessed, noting that glucagon response to food is exaggerated in type 1 diabetes[47]. Additionally, we could not determine if the insulin dose reduction was predominantly meal-time insulin, or exclude that there were inherent differences in the group randomized to metformin, who had a lower insulin dose requirement at baseline. However we were reassured by the consistency of our findings relative to other studies reporting reduced insulin dose requirements[14].

Multi-tissue insulin resistance is a prominent metabolic defect in adults with type 1 diabetes and can occur in the absence of factors that typify the metabolic syndrome. However, the addition of metformin to insulin for 6 months did not improve insulin resistance. Of the secondary endpoints, the findings favoured metformin for insulin dose reduction. These results do not support prescribing metformin to reduce insulin resistance in adults with type 1 diabetes but suggest that metformin may reduce insulin dose via mechanisms independent of insulin resistance. The potential cardiovascular benefits of insulin sparing and reducing peripheral hyperinsulinemia warrant further study.

## Methods
### Study design
INTIMET was a single-site investigator-initiated phase 3 randomized placebo-controlled trial that was approved by the St Vincent's Hospital Human Research and Ethics Committee (Sydney) and was conducted in accordance with the principles of the Declaration of Helsinki. The trial was prospectively registered on 17th October 2019 prior to the start of participant enrolment (Australian New Zealand Clinical Trials Registry, ACTRN12619001440112). The trial did not involve a data safety monitoring board. Participants provided written informed consent and were not compensated. The trial protocol was previously published, and discrepancies from the study as original planned are as stated[48]. Assessment of energy expenditure using indirect calorimetry was planned for in our original protocol but was aborted during the COVID-19 pandemic due to concerns regarding adequate infection control. Therefore, energy expenditure and oxidative metabolism data were not available.

**Participants.** Participants were enroled between November 2019 and December 2021. All metabolic measurements occurred within the Clinical Research Facility (CRF) of the Garvan Institute of Medical Research, Darlinghurst, New South Wales Australia, and data were collected, recorded and stored within the Garvan REDCap database. The study was conducted in 40 participants with type 1 diabetes who met the following criteria, and 20 participants without diabetes were studied at baseline only.

## Study eligibility criteria
Inclusion
1. Age 20–55 years.
2. Pre-menopausal.
      And if type 1 diabetes
3. At least 10 years since diagnosis of type 1 diabetes.
4. Fasting c-peptide <0.3 nmol/L.
5. HbA1c ≤ 9.5% (≤80 mmol/mol).

   Exclusion
1. Current smoking.
2. Medications affecting glucose metabolism (including, but not limited to glucocorticoids, anti-psychotic medications, immuno-suppressants, glucagon-like peptide-1 receptor agonists, sodium-glucose cotransporter inhibitors).
3. Exposure to metformin within the last 30 days (a wash out period was permitted).
4. Alcohol intake greater than 40 g/day in men or greater than 20 g/day in women
5. History of significant weight change (defined as greater than 5%) in the last 3 months, or history of weight loss surgery.
6. Pregnancy or breastfeeding.
7. Major organ dysfunction (including eGFR <60 ml/min/1.73 m², liver disease transaminases greater than 3 times the upper limit of normal, cardiac event within the last 6 months, current cancer or uncontrolled thyroid dysfunction).
8. Diabetic ketoacidosis or severe hypoglycemia (defined as hypoglycemia requiring third-party assistance) in the last 6 months.

## Intervention
After completing baseline studies, participants with type 1 diabetes were randomized to metformin extended release, or identical appearing placebo, as 500 mg tablets, titrated from 500 mg for 7 days, then 1000 mg for 7 days then 1500 mg for the remaining 26-week treatment period. Dosage could be adjusted to accommodate adverse effects, and a submaximal dose was permitted. Pill count was used to assess medication adherence. Randomization, performed by an independent clinician, was 1:1 for metformin to placebo using a computerized minimization procedure to stratify for BMI, HbA1c, sex and age (40%, 20%, 20% and 20% weighting, respectively)[49]. Sex was determined by self-report at enrolment. Participants, outcome assessors and data analysts were blinded to treatment allocation.

## Study procedures
At baseline for all participants with and without diabetes, and at 26 weeks for randomized participants with type 1 diabetes, comprehensive metabolic assessments were performed including blood pressure, BMI and waist-to-hip ratio measurement. Body composition was assessed using dual-energy x-ray absorptiometry (Lunar Prodigy GE-Lunar, CoreScan™ algorithm), to determine fat and fat-free mass (FFM) and visceral adipose tissue (VAT) mass, Magnetic Resonance Imaging (MRI) (GE Architect 3.0-T, GE Healthcare, software version DV28 r5, Chicago, Illinois, image analysis by AMRA® Researcher AMRA Medical AB, Linköping), and transient elastography (Fibroscan® 502 Touch, FibroScan Expert 630®, Echosens, Paris France) to quantify liver and muscle fat. We assessed arterial stiffness using radial artery

applanation tonometry fasted and during the basal period of the clamp procedure during euglycemia (SphygmoCor., AtCor Medical, Australia) to compute the augmentation index (AIx), a validated surrogate of cardiovascular disease[19].

In participants with type 1 diabetes, insulin doses were recorded for up to 14 days in the pre-treatment and end of treatment period. Records were kept manually by participants treated with insulin injections or downloaded using manufacturer-specific software if treated with an insulin pump. Participants with type 1 diabetes wore blinded continuous glucose monitoring (CGM). The first 20 enrolled participants were provided the Medtronic iPro ® Enlite system, (changed after 7 days, in total 2 devices worn per participant) (Medtronic Northridge, CA) and the final twenty participants were provided the Dexcom G6 monitor (Dexcom, San Diego, CA) due to discontinued supply of the Medtronic system mid-trial. These systems were inserted subcutaneously as per manufacturer specifications to measure interstitial glucose, providing measurements approximately every 5 minutes. The recorded CGM outcomes measures included mean sensor glucose, % time in range, % time <3.0 mmol/L, % time <3.9 mmol/L, % time >10 mmol/L, % time >13.9 mmol/L, glucose variability (coefficient of variation; CV%, defined as standard deviation glucose divided by mean glucose*100), mean overnight (0000 – 0559AM) and mean daytime (0600-2359PM) glucose.

To compute these metrics, valid time-stamped sensor glucose data were manually sorted according to glucose value and time of day. At least valid 3 days of CGM data per sensor period were required for analysis. A 24-h period was defined as 0000–2359 h. A day was defined as eligible if ≤20% of the datapoints were missing during that period. Missing sensor data were not imputed.

Diet records were collected for up to 14 days (Easy Diet Diary, Xyris. Brisbane Australia) and analysed using Foodworks 10 Professional (Xyris Pty Ltd, 2019). Retinal photographs were graded using the International Clinical Diabetic Retinopathy (ICDR) severity scale[50]. Urine albumin-creatinine ratio (ACR) data was collated from the preceding 12 months and albuminuria was coded as present or absent based on the single sample.

Participants withheld caffeine, alcohol and strenuous physical activity for 48 h prior to their planned clamp procedure. Women were studied in the follicular phase of the menstrual cycle. Participants with type 1 diabetes underwent washout of their subcutaneous insulin under the supervision of the study Endocrinologist, per published study protocol[48].

Participants arrived fasted for a 6-h, three-stage hyperinsulinemic-euglycemic clamp[48]. The first stage (basal-phase) was a 2-h tracer equilibration period (6,6-²H₂-glucose, Cambridge Isotope Laboratories), started by a priming bolus (5 mg/kg; administered over 5 min) then continuous infusion of the 6,6-²H₂- glucose (Cambridge Isotope Laboratories) (0.05 mg/kg/min), which was continued for at least 2 h, aiming for a 2% enrichment by the time of the steady state basal sampling phase. Participants with type 1 diabetes also received a variable insulin infusion (Actrapid, Novo Nordisk) to achieve a target glucose of 5.5 mmol/L during subcutaneous insulin washout[48]. In the second stage, a fixed insulin infusion was administered at 20 mU/m²/min for 2 h (low-dose phase), with reduction of the tracer infusion dose to 0.025 mg/kg/min. The final high-dose phase was initiated with an insulin bolus (240 mU/m²/min over 2 min, then 120 mU/m²/min over 2 min), then continued at 60 mU/m²/min for 2 h. Body surface area (BSA) was calculated according to the Du Bois formula[51].

Hepatic and adipose insulin sensitivity were defined by EGP and NEFA levels respectively, after partial suppression by insulin during the steady state of the low-dose phase. EGP was calculated using Steele non–steady-state equations, modified for use with stable isotope-labelled glucose[52]. We used a spiked dextrose infusion (25% dextrose with 2% 6,6-²H₂-glucose enrichment) to maintain euglycemia and stable tracer enrichment during the low- and high-dose phases. Muscle

insulin sensitivity was defined as the high-dose phase GIR required to achieve the target glucose level. EGP and GIR data were normalized to fat-free mass. For clamp analysis, we took arterialized samples once steady-state conditions were achieved. Steady-state was defined as minimally fluctuating glucose levels for unchanging GIR and occurred within the last 30 min of each clamp phase. Glucose tracer enrichment was analysed via positive chemical ionization gas chromatography-mass spectrometry using the glucose methyloxime pentaproprionate derivatization strategy, per published methods[53].

Fasting plasma and serum samples were collected for metabolite assessment on pre- and post-treatment clamp study days prior to clamp commencement, including serum for GDF15 (analysed using an in-house ELISA method per previous descriptions), insulin, NEFA and SHBG, and plasma for other blood measures including lipids, uric acid, LFT, IGF-1, adiponectin, IL-6, sE-selectin, sICAM-1 and glucagon[54,55]. Any samples below the sensitivity of the assay were assigned a value equivalent to the limit of quantification (LoQ) divided by the square root of 2 (LoQ/√2).

### Trial endpoints
The primary endpoint was change in liver insulin sensitivity after 26 weeks. Pre-specified secondary endpoints included the change in muscle insulin sensitivity, change in adipose insulin sensitivity, insulin dose, HbA1c and continuous glucose monitoring parameters, GDF15 and other vascular and metabolic measurements (reported in Supplementary Table S4). Glucagon analyses were exploratory. Microbiome analyses are ongoing and not reported in this article. Liver fibrosis measures were not reported, as all values fell within the lower normal reference range in our cohort. CRP analysis was not performed as the available assay did not meet the sensitivity requirements for this study.

### Statistics and reproducibility
**Randomized control trial.** The primary endpoint was assessed by the time-by-treatment group interaction using a generalized linear mixed model with low-dose EGP at baseline and 26 weeks as the response variable. Group allocation, time, and group-by-time interaction terms were included as fixed effects, and a random intercept was assigned for each participant. If a baseline difference was present, a group-by-time interaction model without treatment group as a fixed effect was used[56]. All analyses were adjusted for baseline value. Model distributions and link were selected based on residual plots.

In the initial protocol, basal Ra was listed with low-dose EGP as a co-primary endpoint. Basal Ra was removed after trial commencement but before trial completion due to concerns regarding the variable insulin infusion during the basal clamp period impacting the validity of the basal Ra measure. Since participants with type 1 diabetes tended to arrive with glucose levels above the glucose target, variable insulin concentrations were infused in the type 1 diabetes participants during the basal phase to achieve euglycemia. Higher insulin concentrations would then excessively suppress Ra in the type 1 diabetes participants compared to control participants, and variable insulin concentrations during the basal phase would invalidate the basal Ra measures in type 1 diabetes as each participant would be subjected to a different insulin concentration depending on their arrival glucose. Further, insulin concentrations measured in type 1 diabetes represent exogenously infused insulin into the peripheral circulation. Unlike type 1 diabetes, in control participants without diabetes, plasma insulin concentrations measured during the basal phase represent endogenous insulin secreted from the pancreas to the portal circulation, with hepatic clearance before reaching the peripheral circulation. Thus, the basal phase served the purpose of tracer equilibration, and achievement of euglycemia only.

To detect an end-of-treatment difference between groups in EGP of 0.3 mg/kg/min with 80% power (presumed 0.3 mg/kg/min standard deviation; equivalent to 2.4 μmol/kgFFM/min, alpha level 0.05), we needed 17 participants in each group, increased to 20 to accommodate up to 15% participant attrition.

Data were analysed by intention-to-treat principles without imputation. All assessments used two-sided tests and a p-value ≤ 0.05 was considered statistically significant. For secondary analyses, a Bonferroni corrected p-value was applied to account for multiple comparisons. Normally distributed data were presented as mean and standard deviation (SD) and analysed using parametric tests. Skewed data were presented as median and interquartile range (IQR) then analysed with non-parametric tests or as log-transformed data analysed with parametric tests. Categorical data were presented as counts and percentages. We did not perform specific sex-based analyses, as the study was not powered for subgroup analyses.

**Baseline cross-sectional study.** Two-sample t-tests, Mann–Whitney U or Chi-square tests were used to compare baseline characteristics between participants with and without type 1 diabetes. We determined relationships between insulin resistance and cardiometabolic variables using Pearson, Spearman's, and regression models to determine predictors of insulin resistance. As exploratory analyses, these data are presented without correction for multiple testing.

**Excluded data points.** A single outlying GDF15 value from the type 1 diabetes group was excluded from the analysis. This datapoint was from an individual who was subsequently diagnosed with a medical condition deemed likely to falsely elevate serum GDF15 levels. Hormonal contraceptive users were excluded for all analyses involving SHBG.

Statistical analyses were performed using SPSS version 28.0.1.0 (IBM Corp. Released 2021. IBM SPSS Statistics for Macintosh, Version 28.0. Armonk, NY: IBM Corp). Due to the nature of the data, Rstudio (Rstudio 2023.06.1 + 524 "Mountain Hydrangea") using the glmmTMB package was used for continuous glucose monitoring data as SPSS did not have the beta-binomial distribution option within its generalized linear mixed modelling framework. A p-value ≤ 0.05 was considered statistically significant.

### Reporting summary
Further information on research design is available in the Nature Portfolio Reporting Summary linked to this article.

## Data availability
The data generated in this study has not been deposited in a public repository due to absence of consent from study participants, and restrictions placed by our study's ethics approval. Individual participant data (including data dictionaries) that underlie the results reported in this article are available after deidentification. Any proposed use of data requires approval by an independent review committee (St Vincent's Hospital Human Research and Ethics Committee [Sydney]) and will be provided under a data sharing agreement. Requests for data should be directed to the corresponding author within 5 years of study publication. The corresponding author will respond within 2–4 weeks of receipt.

## Code availability
No custom computer code or algorithm was used to generate the analysis.

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

## Acknowledgements

This study was an initiative of the Australian Collaborative Towards Adjunctive Therapies in Type 1 Diabetes (ACT-T1D) Study Group. We thank Timothy Connor, Lijun Zhao, Leonie Heilbronn, Gary Wittert, Hong-Ping Zhang, Vicky Tsai, Andrzej Januszewski and Alicia Jenkins for their assistance in processing metabolite samples; Hamish Dunn for grading retinal photographs, the staff at UNSW Stats Central for supporting study design and analysis; Alex Viardot for participant randomization, Mark Danta, Rebecca Hickey and Yael Barnett for imaging support; Krisztina Toth and Renee Richens from the Garvan Clinical Research Facility for assistance in conducting the studies; our funders for supporting the project; and finally, the study participants for their generous participation. INTIMET was supported by grants from the Diabetes Australia Research Program (J.R.S. and J.R.G.; RG182876/Y19M1-GREEJ), St Vincent's Clinic Foundation (J.R.G.), University of New South Wales Cardiac Vascular and Metabolic Medicine Theme (J.R.S.), National Health and Medical Research Council (NHMRC) (J.R.S.; GNT1189721), and philanthropic donations (Jonathon and Melissa Green; J.R.G.). Funders did not have a role in the design or interpretation of the study.

## Author contributions

J.R.S., N.O., J.E., G.M.K., C.R.B., D.S.B., D.J.H.W. and J.R.G. designed the study. J.R.S., J.E. and J.R.G. conducted the study, and D.J.H.W. identified participants for the study. G.M.K. and C.R.B. performed tracer analysis. J.R.S. and N.O. carried out the data and statistical analysis. S.B. conducted GDF15 and glucagon analyses. D.S.B. conducted dietary analyses. All authors critically reviewed the manuscript. J.R.S. affirms to the honest, accurate, and transparent account of the reported research and that no important or relevant aspects of the study have been omitted.

## Competing interests

The authors declare no competing interests.
