## [Transparent Peer Review file · Nature Communications]

Effect of Metformin on Insulin Resistance in Adults with Type 1 Diabetes: a 26-week Randomized Double-blind Clinical Trial

Corresponding Author: Dr Jennifer Snaith

Version 0:

Reviewer comments:

Reviewer #1

(Remarks to the Author)

Thanks to the authors for thoughtful replies to the reviewer comment - this is a challenging area of investigation.

- 1) The authors appear to accept in their comments that between study protocol differences could potentially account for the differences between their finding in adults and the finding of the earlier trial in adolescents - this should also be included in the manuscript
- 2) The use of reference 35 in the Discussion does not acknowledge that the estimated increase in CVD in association with higher insulin dose appeared to be mediated by other CV risk factors.
- 3) I still find the bar charts and error bars a suboptimal method for showing the data (although this was not an issue for other reviewers).

Reviewer #2

(Remarks to the Author)

1. Original comment: "Clamp procedure: It seems to me that what is authors call "basal phase" is endogenous glucose production assessed after insulin infusion into a peripheral vein to obtain normoglycemia in the patients with diabetes. It is also stated that basal rate of appearance of glucose was removed as a co-primary end-point due to concerns regarding the variable insulin infusion during the basal clamp period. Please explain how you calculated EGP without Ra? How much insulin was given to achieve normoglycemia, compared to the 20 mU/m²/min maintained for 2h during the second stage of the clamp? Please discuss this when comparing to non-diabetic controls that were assessed during their habitual fasting insulin levels that of course comes from pancreatic secretion into the portal system"
 - a. Latter paragraph: Please include a comment in the Discussion about the differences in evaluation of insulin levels measured in the peripheral blood that comes from exogeneous insulin infusion (patients) vs insulin secreted from the pancreas/pre-hepatic
2. Original comment; "Insulin resistance measured during high dose insulin infusion was given as GIR. Was that corrected for tracer-estimated endogenous glucose production?"
 - a. Thanks for the explanation. However, you fail to show/substantiate that endogenous glucose production was suppressed in all individuals during the high dose clamp. Please comment/include as a limitation.
3. Original comment: "A major determinant of insulin resistance in patients with type 1 diabetes is believed to be the degree of hyperglycemia ("tissue glucose toxicity"). It is not mentioned/discussed at all by the authors? Table S2 shows a significant correlation between 1/GIR and HbA1c. Please also show correlation to fasting glucose. Were differences in insulin resistance between patients and control still significant after adjustment for HbA1c and/or fasting glucose?"
 - a. Thank you for providing supplementary data and comment. However, although long term glucose levels, as proxied by HbA1c may be important for insulin sensitivity in type 1 diabetes, actual glucose levels as reflected in CGM-data 24 h before the clamp or fasting glucose on the clamp-day may be at least as important. Please comment?

Reviewer #4

(Remarks to the Author)

Snaith et al report on a mixed cross-sectional study of adults with Type 1 diabetes vs. controls, and a randomised, placebo-controlled trial of metformin in the group with Type 1 diabetes. This review is unusual in that the remit was to assess whether the authors had responded to the original (statistical) review of the paper.

Looking at the first comment of Reviewer 3:

1) The study used repeated measure model to estimate treatment effect and used a model without the treatment main effect term when baseline values were different between groups, citing Twisk 2018 (ref 10). However, Twisk advised always to adjust baseline value regardless of whether it was statistically significant between groups. Twisk 2018 also showed that longitudinal analysis of covariance or analysis of changes adjusting baseline value were equivalent. The repeated measure analysis without treatment main effect is advantaged when there is missing data in follow up, which is not the case here. It is therefore advised for this study to use longitudinal analysis of covariance or analysis of changes adjusting baseline for all continuous outcomes (primary or secondary).

Based on the authors' response, however, it appears that they have changed the analysis.

When the data consist of a continuous measure at baseline and follow-up, there are two methods that make sense, as outlined in the paper by Twisk. The most common approach is to fit a model with the follow-up measurement as the response, and two predictors: the treatment variable, and the baseline measurement of the outcome variable. Equivalently, the response in the model can be the change from baseline to follow-up; the predictors are still the treatment variable and the baseline measurement of the outcome. Both of these approaches give the same result for the treatment effect.

An alternative approach is to fit a repeated measures model, in which all measurements of the outcome variable, both from baseline and from follow-up, are treated as part of the response variable. The (fixed-effect) predictors in the model are a binary variable for time, and a treatment-by-time interaction. The model includes a random effect for participants. There is no fixed effect for treatment, since this would imply a between-group difference at baseline, but since the study is randomised, it is known that any observed difference at baseline is due to chance, and this is captured in the model by the random participant effects.

Twisk recommends either approach as being acceptable. It suggests that a repeated measures model including a fixed effect for treatment should not be used, but it is this model that has been used by the authors for most analyses. They use a model without a fixed effect for treatment only when they observe a between group difference at baseline, which is incorrect.

So, in short, the authors should either use the repeated measures model without a fixed effect for treatment (for all analyses), or use the simpler approach of ANCOVA.

Reviewer 3's second and third comments are not about statistics, but their fourth comment asked for Bonferroni adjustment. The authors push back against this suggestion since the analyses in question are exploratory. I fully support this response. Responses to the reviewers fifth and sixth comments are also satisfactory.

The reviewer's seventh comment is odd. Trials are powered to detect an a priori treatment effect; they are not supposed to be powered to detect the treatment effect that is actually observed (e.g. this might be zero). I also disagree with the comment about needing power calculations for secondary outcomes. The authors rightly rebut these comments. However, I did note that the power calculations assumed the standard deviation of the primary outcome to be 0.3 mg/kg/min. On reading this, I realized that the paper does not include summaries of the outcomes. I would expect to see summaries (i.e. mean and SD, or median and IQR) for each outcome at baseline and at follow-up, and ideally for the change over time.

The nearest I could find was in the supplements, table S4. Here we get the mean at baseline, at 26w, and for the change, but instead of SDs, we are given 95% CIs. I would much rather see the SDs. Anyway, from the CIs for the primary outcome I calculated the SD for the change from baseline to 26w to be approximately 2.6 mg/kg/min. Maybe I made a mistake, but if this is true, then the study was in fact severely underpowered, since the original calculation assumed the SD to be 0.3. This needs to be clarified: the paper should report summaries of the outcomes at each time point, and the changes from baseline, and if the SD of the primary outcome was much larger than originally assumed, then the lack of power to detect the target treatment effect should be noted as a limitation. Note, given the CI for the treatment effect of -1.6 to +1.9, the original treatment effect of 0.3 that the study was meant to be powered to detect is very much consistent with the data. In other words, the study does not by any means rule out a treatment effect of 0.3 mg/kg/min, or quite a bit larger than that. In that respect, the trial is inconclusive.

The other responses to Reviewer 3's comments are straightforward.

The following comments are my own, though some were stimulated by other reviewers' comments.

In Figure 1, the error bars should show 95% confidence intervals, not SEM.

In Figure 3, the error bars do show 95% CIs, which is good, but the use of bars to indicate the mean value is not so good. I would prefer a graph with time on the x axis, symbols for the mean values, and 95% CIs as error bars. I.e. a similar approach to Figure 1. This should not take long to do.

Table S3 is insufficient. In the model for GIR, it may be that adding HbA1c does not improve the fit of the model very much, but I would like to see the regression coefficients and CIs (and R-squared values) from three models: (1) GIR ~ T1D; (2) GIR ~ HbA1c; and (3) GIR ~ T1D + HbA1c. If HbA1c is a better predictor of GIR than T1D, and if the coefficient for T1D is much smaller in model (3) than model (1), then I think that tells a very different story. If going down this path, then the authors need to report these models in more detail; R-squared statistics alone are not enough. Note, the statement “However, using multivariate testing, HbA1c did not significantly alter the relationship between type 1 diabetes and insulin resistance in our cohort” is not supported by Table S3; one would need to see the regression coefficients from models (1) and (3), as described above, in order to make this statement.

The primary analysis confidence interval in the abstract is misquoted. It should be -1.6 to +1.9 (not 1.0).

This is a bug-bear of mine: with 40 people with T1D, and 20 controls, how can the controls be said to be “matched” to the people with T1D? Maybe they were selected to have similar characteristics, but with different numbers of people, they cannot be said to be matched.

There is a statement “Therefore, adults with and without type 1 diabetes did not differ in age, sex or anthropometric characteristics (Table 1).” However, no data on sex are reported in this paragraph.

It seems odd to report results from regression models fitted on a log scale, but report mean changes and CIs for data on the original scale. As stated elsewhere, I would rather see descriptive summaries of the changes over time; if the data are not normal, then median and IQR could be used for these summaries.

Reviewer #5

(Remarks to the Author)

Version 1:

Reviewer comments:

Reviewer #1

(Remarks to the Author)

No further comments.

Reviewer #2

(Remarks to the Author)

The authors have adequately addressed my concerns and corrected the manuscript accordingly.

Reviewer #4

(Remarks to the Author)

Re Snaith et al revised paper.

I thank the authors for their responses to my comments, and by and large I am happy.

I apologise for not spotting the different units used in the sample size calculation and the reporting of the results. Perhaps this needs to be highlighted in the sample size section of the paper, to avoid confusion? Perhaps also add a note to say what a SD of 0.3 mg/kg/min would equate to (roughly) in $\mu\text{mol/kgFFM/min}$?

I still think my comments about reporting the data still stand. Table S4 is reporting means and CIs, but this is slightly misleading – the footnote states “Data are mean (95% CI)” but then goes on to say that they are predicted values from the models. Perhaps the table title could be clearer in saying that the table shows model-predicted mean values and treatment effect estimates? Anyway, what I am asking for is summaries (mean and SD, or median and IQR) of the actual data, not predictions from the models. I think this should always be given somewhere in a paper. I find it odd to look at a paper that does not present summaries of the study outcomes, as measured.

This could be done within Table S4, by preplacing the model-predicted mean and 95% CI with the actual measured mean and SD (or median and IQR, if data are skewed). The model-predicted changes over time within groups, and the treatment effect estimates, could stay as they are.

Alternatively, another table could be added to present summaries of the data by treatment group and time point.

Reviewer #5

(Remarks to the Author)

REVIEWER COMMENTS

Changes made outside of reviewer comments:

We have included an additional recently published reference supporting a role for the effect of metformin and the gut. This study reported evidence for direct transit of glucose from the circulation to the jejunum in metformin treated individuals, and supports our discussion of possible non-insulin resistance mediated effects of metformin.

New reference: Sakaguchi K, Sugawara K, Hosokawa Y, et al. Metformin-regulated glucose flux from the circulation to the intestinal lumen. *Commun Med* 2025;5(1):44.

Reviewer #1 (Remarks to the Author):

Thanks to the authors for thoughtful replies to the reviewer comment - this is a challenging area of investigation.

1) The authors appear to accept in their comments that between study protocol differences could potentially account for the differences between their finding in adults and the finding of the earlier trial in adolescents - this should also be included in the manuscript.

We acknowledge the variability in clamp and tracer methodology, and that our approach differed from the approach taken in adolescent studies. The limitation section of the discussion now acknowledges this (new text underlined):

‘Our study may have been underpowered to detect the effect of metformin on GIR, or other secondary outcomes, and differences in clamp methodology may account for differences compared to findings reported in adolescents.^{15,16}’

2) The use of reference 35 in the Discussion does not acknowledge that the estimated increase in CVD in association with higher insulin dose appeared to be mediated by other CV risk factors.

Thank you for the helpful comment, encouraging a nuanced interpretation of the referenced study. We have revised the sentence to clarify that the association between insulin dose and CVD risk may be mediated by other risk factors. The text now reads (new text underlined):

‘Notably, in a Diabetes Control and Complications Trial/Epidemiology of Diabetes Interventions and Complications (DCCT/EDIC) follow up study, higher insulin doses were adversely associated with cardiovascular risk factors, and each 0.1 units/kg/day insulin dose increase predicted a 6% increase in cardiovascular disease risk over 30 years. Although the association with cardiovascular outcomes was attenuated after adjustment for traditional risk factors, the findings raise the hypothesis that minimising insulin dose may have beneficial consequences for cardiovascular risk over time.’

3) I still find the bar charts and error bars a suboptimal method for showing the data (although this was not an issue for other reviewers).

As requested, the GIR, EGP and NEFA subpanels in Figure 3 now mirror the time-series format used in figure 1. The metformin and placebo groups have been presented side by side, as the data would be obscured by overlaying 4 charts. There are also presented with 95% CI.

Reviewer #2 (Remarks to the Author):

1. Original comment:

“Clamp procedure: It seems to me that what is authors call “basal phase” is endogenous glucose production assessed after insulin infusion into a peripheral vein to obtain normoglycemia in the patients with diabetes. It is also stated that basal rate of appearance of glucose was removed as a co-primary end-point due to concerns regarding the variable insulin infusion during the basal clamp period. Please explain how you calculated EGP without Ra? How much insulin was given to achieve normoglycemia, compared to the 20 mU/m²/min maintained for 2h during the second stage of the clamp? Please discuss this when comparing to non-diabetic controls that were assessed during their habitual fasting insulin levels that of course comes from pancreatic secretion into the portal system”

a. Latter paragraph:

Please include a comment in the Discussion about the differences in evaluation of insulin levels measured in the peripheral blood that comes from exogeneous insulin infusion (patients) vs insulin secreted from the pancreas/pre-hepatic

This point is an important distinction and would be of benefit to the Nature Communications readership to clarify further. Although it was suggested that we add text to the discussion, for flow and clarity we elected to add the extra text to the corresponding section of the methods section. We feel that this addition improves both the transparency in refining the primary endpoint to remove basal Ra and contextualises physiological differences between individuals with and without type 1 diabetes. Thank you to the reviewer for this suggestion.

The text added to the methods section:

‘Since participants with type 1 diabetes tended to arrive with glucose levels above the glucose target, variable insulin concentrations were infused in the type 1 diabetes participants during the basal phase to achieve euglycemia. Higher insulin concentrations would then excessively suppress Ra in the type 1 diabetes participants compared to control participants, and variable insulin concentrations during the basal phase would invalidate the basal Ra measures in type 1 diabetes as each participant would be subjected to a different insulin concentration depending on their arrival glucose. Further, insulin concentrations measured in type 1 diabetes represent exogenously infused insulin into the peripheral circulation. Unlike type 1 diabetes, in

control participants without diabetes, plasma insulin concentrations measured during the basal phase represent endogenous insulin secreted from the pancreas to the portal circulation, with hepatic clearance before reaching the peripheral circulation. Thus, the basal phase served the purpose of tracer equilibration, and achievement of euglycemia only.'

2. Original comment; "Insulin resistance measured during high dose insulin infusion was given as GIR. Was that corrected for tracer-estimated endogenous glucose production?"

a. Thanks for the explanation. However, you fail to show/substantiate that endogenous glucose production was suppressed in all individuals during the high dose clamp. Please comment/include as a limitation.

Indeed, we did not correct for EGP in our insulin resistance measures during the high-dose phase. The high dose clamp EGP levels in participants with and without diabetes are demonstrated in Figure 1D and exhibit complete or near complete suppression of EGP during this phase, and there was no difference between high-dose EGP in participants with or without diabetes, or between placebo or metformin groups at baseline or 26 weeks (Figure 1D and Figure 3).

The following text has been added to the discussion section for clarity:

'Although EGP was almost completely suppressed in most participants, we did not correct for tracer estimated EGP in the high-dose phase. This may have led to an underestimation of GIR in some participants, however there were no observable differences in between group EGP between participants with and without diabetes, or between placebo and metformin groups.'

3. Original comment:

"A major determinant of insulin resistance in patients with type 1 diabetes is believed to be the degree of hyperglycemia ("tissue glucose toxicity"). It is not mentioned/discussed at all by the authors? Table S2 shows a significant correlation between 1/GIR and HbA1c. Please also show correlation to fasting glucose. Were differences in insulin resistance between patients and control still significant after adjustment for HbA1c and/or fasting glucose?"

a. Thank you for providing supplementary data and comment. However, although long term glucose levels, as proxied by HbA1c may be important for insulin sensitivity in type 1 diabetes, actual glucose levels as reflected in CGM-data 24 h before the clamp or fasting glucose on the clamp-day may be at least as important. Please comment?

As a measure of recent glucose status, we now report the fasting glucose level on the day of the clamp, correlated with GIR and EGP in participants with type 1

diabetes. We found no significant relationship between either measure of insulin resistance and fasting glucose. This data has been added to Table S2.

Given the correlation between GIR and HbA1c, we now present in further detail an analysis of the effect of acute and chronic glycemia on GIR. For clearer reporting, we elected to replace the stepwise regression analysis, with generalized linear modelling. We present adjustment for fasting glucose (defined by glucose level at the time of commencement of the clamp) and HbA1c separately. We found that fasting glucose did not significantly alter the difference in GIR between participants with and without type 1 diabetes. When examined separately, adjusting for HbA1c attenuated the between group difference, suggesting a possible effect of chronic hyperglycemia on GIR.

This is presented in Table S3, displaying beta coefficients, standard errors and p-values for each parameter in predicting GIR. The manuscript has been adjusted accordingly.

In the section assessing the relationship between metabolic characteristics and insulin resistance, the following text has been added (new text underlined):

In participants with type 1 diabetes, muscle insulin resistance additionally correlated directly with HbA1c, insulin dose, mean overnight glucose, and inversely with glycemic variability (Supplementary Table S2). Adjustment for HbA1c, but not fasting glucose, attenuated the difference in GIR between participants with and without type 1 diabetes (Supplementary Table S3).

In the discussion section, the following text has been updated (new text underlined):

Our study suggests that in type 1 diabetes, muscle insulin resistance is influenced by hyperglycaemia. It is also likely induced by the direct absorption of subcutaneously administered insulin into the peripheral circulation, bypassing hepatic insulin clearance and distorting the usual balance between portal and peripheral insulin concentrations.

AND

Our trial could have provided different results if targeted to a cohort with a higher BMI, or higher HbA1c. Our enrolment criteria excluded those with HbA1c greater than 9.5%, a population in whom glucotoxicity may significantly contribute to insulin resistance. However, even with this restriction, HbA1c appeared to predict insulin resistance in our cohort.

Reviewer #4 (Remarks to the Author):

Snaith et al report on a mixed cross-sectional study of adults with Type 1 diabetes vs. controls, and a randomised, placebo-controlled trial of metformin in the group with Type 1 diabetes. This review is unusual in that the remit was to assess whether the authors had responded to the original (statistical)

review of the paper.

Looking at the first comment of Reviewer 3:

1) The study used repeated measure model to estimate treatment effect and used a model without the treatment main effect term when baseline values were different between groups, citing Twisk 2018 (ref 10). However, Twisk advised always to adjust baseline value regardless of whether it was statistically significant between groups. Twisk 2018 also showed that longitudinal analysis of covariance or analysis of changes adjusting baseline value were equivalent.

The repeated measure analysis without treatment main effect is advantaged when there is missing data in follow up, which is not the case here. It is therefore advised for this study to use longitudinal analysis of covariance or analysis of changes adjusting baseline for all continuous outcomes (primary or secondary).

Based on the authors' response, however, it appears that they have changed the analysis.

When the data consist of a continuous measure at baseline and follow-up, there are two methods that make sense, as outlined in the paper by Twisk. The most common approach is to fit a model with the follow-up measurement as the response, and two predictors: the treatment variable, and the baseline measurement of the outcome variable. Equivalently, the response in the model can be the change from baseline to follow-up; the predictors are still the treatment variable and the baseline measurement of the outcome. Both of these approaches give the same result for the treatment effect.

An alternative approach is to fit a repeated measures model, in which all measurements of the outcome variable, both from baseline and from follow-up, are treated as part of the response variable. The (fixed-effect) predictors in the model are a binary variable for time, and a treatment-by-time interaction. The model includes a random effect for participants. There is no fixed effect for treatment, since this would imply a between-group difference at baseline, but since the study is randomised, it is known that any observed difference at baseline is due to chance, and this is captured in the model by the random participant effects.

Twisk recommends either approach as being acceptable. It suggests that a repeated measures model including a fixed effect for treatment should not be used, but it is this model that has been used by the authors for most analyses. They use a model without a fixed effect for treatment only when they observe a between group difference at baseline, which is incorrect.

So, in short, the authors should either use the repeated measures model without a fixed effect for treatment (for all analyses), or use the simpler approach of ANCOVA.

Considering this discussion, we have used a repeated mixed measures model without a fixed effect for treatment group for all analyses, with the individual participant as the random effect. Reflecting this updated analysis, the footnote in table S4 has been updated to remove mention of group as a fixed effect. This approach to the analysis did not change the message, or outcomes reported in this study. However, removing drug as a fixed effect slightly changed the modelled estimates and the affected values have been updated throughout the manuscript.

Reviewer 3's second and third comments are not about statistics, but their fourth comment asked for Bonferroni adjustment. The authors push back against this suggestion since the analyses in question are exploratory. I fully support this response. Responses to the reviewers fifth and sixth comments are also satisfactory.

Thank you for these comments, we have made no further changes.

The reviewer's seventh comment is odd. Trials are powered to detect an a priori treatment effect; they are not supposed to be powered to detect the treatment effect that is actually observed (e.g. this might be zero). I also disagree with the comment about needing power calculations for secondary outcomes. The authors rightly rebut these comments.

Thank you for these comments, we have made no further changes.

However, I did note that the power calculations assumed the standard deviation of the primary outcome to be 0.3 mg/kg/min. On reading this, I realized that the paper does not include summaries of the outcomes. I would expect to see summaries (i.e. mean and SD, or median and IQR) for each outcome at baseline and at follow-up, and ideally for the change over time.

The nearest I could find was in the supplements, table S4. Here we get the mean at baseline, at 26w, and for the change, but instead of SDs, we are given 95% CIs. I would much rather see the SDs. Anyway, from the CIs for the primary outcome I calculated the SD for the change from baseline to 26w to be approximately 2.6 mg/kg/min. Maybe I made a mistake, but if this is true, then the study was in fact severely underpowered, since the original calculation assumed the SD to be 0.3. This needs to be clarified: the paper should report summaries of the outcomes at each time point, and the changes from baseline, and if the SD of the primary outcome was much larger than originally assumed, then the lack of power to detect the target treatment effect should be noted as a limitation. Note, given the CI for the treatment effect of -1.6 to +1.9, the original treatment effect of 0.3 that the study was meant to be powered to detect is very much consistent with the data. In other words, the study does not by any means rule out a treatment effect of 0.3 mg/kg/min, or quite a bit larger than that. In that respect, the trial is inconclusive.

The primary and secondary analyses for our study employed generalized linear mixed modelling (GLMM). When using GLMM, the output typically gives modelled (estimated marginal means), standard errors and 95% confidence intervals. These

come from the fixed and random effects structure assumed in the model, and we are not provided SD, median and IQR from modelled estimates. Thus, we elected to keep the current reporting method in Supplemental Table 4 to report within the limits of the model, though raw baseline data is reported with mean/SD and median/IQR (Table 3).

Nonetheless, concerns were raised about the power of the study to detect a difference in the primary outcome.

Revisiting power -

With the analysis represented without drug as a fixed effect, the primary endpoint (EGP) beta coefficient is 0.2, noting that our previous analysis using drug as a fixed effect provided a more conservative estimate of effect size. The SEM of the primary outcome was 0.3, equating to an SD of 1.8.

The study was powered against a treatment effect of 0.3, and SD 0.3, taken from a study expressing EGP units as mg/kg/min, whereas our EGP units were expressed as $\mu\text{mol/kgFFM/min}$. To provide a direct comparison, we here convert the units (ie. mg to μmol , and from kg body weight to kg fat-free mass).

- Molecular weight of glucose is $180.16\text{g/mol} = 0.18\text{g/mmol}$
- Firstly, to convert glucose from $\mu\text{mol/kgFFM/min}$ to mg/FFMkg/min we multiply by 0.18
 - = $1.8 \mu\text{mol/kgFFM/min} * 0.18$
 - = 0.32mg/kgFFM/min
- Then assuming fat-free mass comprises 70% of body mass, then to convert glucose per unit fat-free mass, to per unit body weight
 - = $0.32 \text{mg/kgFFM/min} * 0.7 = 0.22 \text{mg/kg/min}$

Therefore, with conversion of units, an SD of $1.8 \mu\text{mol/kg/FFM/min} = 0.2 \text{mg/kg/min}$. This is within the assumed SD within which the study was powered and supports that the negative result was true.

The other responses to Reviewer 3's comments are straightforward.

The following comments are my own, though some were stimulated by other reviewers' comments.

In Figure 1, the error bars should show 95% confidence intervals, not SEM.

We thank the reviewer for detecting this. The error bars have been converted to 95% CI.

In Figure 3, the error bars do show 95% CIs, which is good, but the use of bars to indicate the mean value is not so good. I would prefer a graph with time on the x axis, symbols for the mean values, and 95% CIs as error bars. I.e. a similar approach to Figure 1. This should not take long to do.

In the updated Figure 3, time series graphs for multi-tissue measures of insulin resistance have replaced bar chart representations of these measures. Insulin dose and GDF15 data is unchanged, as they are not appropriate for time series charts.

Table S3 is insufficient. In the model for GIR, it may be that adding HbA1c does not improve the fit of the model very much, but I would like to see the regression coefficients and CIs (and R-squared values) from three models: (1) $GIR \sim T1D$; (2) $GIR \sim HbA1c$; and (3) $GIR \sim T1D + HbA1c$. If HbA1c is a better predictor of GIR than T1D, and if the coefficient for T1D is much smaller in model (3) than model (1), then I think that tells a very different story. If going down this path, then the authors need to report these models in more detail; R-squared statistics alone are not enough. Note, the statement “However, using multivariate testing, HbA1c did not significantly alter the relationship between type 1 diabetes and insulin resistance in our cohort” is not supported by Table S3; one would need to see the regression coefficients from models (1) and (3), as described above, in order to make this statement.

Please see response to above regarding adjusting for HbA1c and GIR. We feel generalized linear modelling is a simpler, more transparent method to present the data and have removed the previous modelling.

The primary analysis confidence interval in the abstract is misquoted. It should be -1.6 to +1.9 (not 1.0).

Thank you. The abstract has been updated to reflect the updated treatment difference estimates per models removing treatment group as a fixed effect.

This is a bug-bear of mine: with 40 people with T1D, and 20 controls, how can the controls be said to be “matched” to the people with T1D? Maybe they were selected to have similar characteristics, but with different numbers of people, they cannot be said to be matched.

In the ‘study participants section (page 5), the wording has been changed to Twenty adults without diabetes were recruited for baseline studies only’

There is a statement “Therefore, adults with and without type 1 diabetes did not differ in age, sex or anthropometric characteristics (Table 1).” However, no data on sex are reported in this paragraph.

Thank you for detecting this omission. We have inserted a description of sex distribution. The sentence now reads: ‘The twenty adults without diabetes were aged 37.0 ± 8.4 years, 60% male, with BMI 26.2 ± 4.3 kg/m² and HbA1c $5.1 \pm 0.3\%$. Therefore, adults with and without type 1 diabetes did not differ in age, sex or anthropometric characteristics’

It seems odd to report results from regression models fitted on a log scale, but report mean changes and CIs for data on the original scale. As stated elsewhere, I would rather see descriptive summaries of the changes over time; if the data are not normal, then median and IQR could be used for these summaries.

Thank you - this comment has been likely addressed in our previous response to outcomes reporting. In short, due to limitations in reporting estimates from generalized linear mixed modelling, medians and IQR could not be directly reported.

Reviewer #5 (Remarks to the Author):

REVIEWERS' COMMENTS

Reviewer #1 (Remarks to the Author):

No further comments.

Reviewer #2 (Remarks to the Author):

The authors have adequately addressed my concerns and corrected the manuscript accordingly.

Reviewer #4 (Remarks to the Author):

Re Snaith et al revised paper.

I thank the authors for their responses to my comments, and by and large I am happy.

I apologise for not spotting the different units used in the sample size calculation and the reporting of the results. Perhaps this needs to be highlighted in the sample size section of the paper, to avoid confusion? Perhaps also add a note to say what a SD of 0.3 mg/kg/min would equate to (roughly) in $\mu\text{mol/kgFFM/min}$?

Thank you for this helpful suggestion. The equivalent SD in $\mu\text{mol/kgFFM/min}$ has been added to the sample size section with the methods section of the paper.

I still think my comments about reporting the data still stand. Table S4 is reporting means and CIs, but this is slightly misleading – the footnote states “Data are mean (95% CI)” but then goes on to say that they are predicted values from the models. Perhaps the table title could be clearer in saying that the table shows model-predicted mean values and treatment effect estimates?

Anyway, what I am asking for is summaries (mean and SD, or median and IQR) of the actual data, not predictions from the models. I think this should always be given somewhere in a paper. I find it odd to look at a paper that does not present summaries of the study outcomes, as measured.

This could be done within Table S4, by preplacing the model-predicted mean and 95% CI with the actual measured mean and SD (or median and IQR, if data are skewed). The model-predicted changes over time within groups, and the treatment effect estimates, could stay as they are.

Alternatively, another table could be added to present summaries of the data by treatment group and time point.

The pre and post treatment values in table S4 are now data summaries, replacing model predicted data. The differences between groups, and within

groups remains modelled treatment differences. The table title has been updated to 'The effect of metformin and placebo on primary and secondary metabolic endpoints, from model-predicted treatment estimates', and the column subheadings, also explicitly state that differences are modelled differences.

We have removed the statement 'data are mean (95% CI)' from the footnote of table S4. The footnote now reads 'Baseline and 26-week data are descriptive summaries of the raw data, expressed as mean \pm SD or median (interquartile range). Treatment difference values are modelled estimated means and 95% CI from the intention-to-treat population.'

Reviewer #5 (Remarks to the Author):
